# The NMDA receptor antagonist ketamine impairs and delays context-dependent decision making in the parietal cortex

Yuki Suda[1,2,3] & Takanori Uka [1,2,3 ✉]

Flexible decision making is an indispensable ability for humans. A subanesthetic dose of ketamine, an N-methyl-D-aspartate receptor antagonist, impairs this flexibility in a manner that is similar to patients with schizophrenia; however how it affects neural processes related to decision making remains unclear. Here, we report that ketamine administration impairs neural processing related to context-dependent decision making, and delays the onset of decision making. We recorded single unit activity in the lateral intraparietal area (LIP) while monkeys switched between a direction-discrimination task and a depth-discrimination task. Ketamine impaired choice accuracy for incongruent stimuli that required different decisions depending on the task, for the direction-discrimination task. Neural sensitivity to irrelevant depth information increased with ketamine during direction discrimination in LIP, indicating impaired processing of irrelevant information. Furthermore, the onset of decision-related neural activity was delayed in conjunction with an increased reaction time irrespective of task and stimulus congruency. Neural sensitivity and response onset of the middle temporal area (MT) were not modulated by ketamine, indicating that ketamine worked on neural decision processes downstream of MT. These results suggest that ketamine administration may impair what information to process and when to process it for the purpose of achieving flexible decision making.

[1] Department of Integrative Physiology, Graduate School of Medicine, University of Yamanashi, 1110 Shimokato, Chuo, Yamanashi 409-3898, Japan. [2] Brain Science Institute, Tamagawa University, 6-1-1 Tamagawagakuen, Machida, Tokyo 194-8610, Japan. [3] Department of Neurophysiology, Graduate School of Medicine, Juntendo University, 2-1-1 Hongo, Bunkyo, Tokyo 113-8421, Japan. ✉email: tuka@yamanashi.ac.jp

Our daily life is full of flexible behavior, allowing us to immediately select optimal behavior among many options depending on the context. This ability relies on flexible decision making, which has been examined by researchers using the task-switching paradigm[1]. In this paradigm, subjects are presented with a stimulus that involves at least two different features, and the subject chooses the optimal option according to the task they should perform. Therefore, this paradigm requires the ability to decode the relevant feature and discard irrelevant information. This ability is particularly common in humans[2] and non-human primates[3]; however, it is impaired in patients with schizophrenia[4]. Notably, systemic administration of ketamine, an N-methyl-d-aspartate (NMDA) receptor antagonist, impairs the switching performance of monkeys[5] in a manner similar to patients with schizophrenia[6], suggesting that neural processing mediated by NMDA receptors plays a crucial role in flexible decision making.

To address how ketamine administration is related to flexible decision making, we focused on neural activity in the lateral intraparietal (LIP) area, a cortical area that contains decision-related neural activity, using a cued task-switching paradigm that employed perceptual decision making. In perceptual decision making, LIP neurons demonstrate build-up activity to sensory stimuli that peaks just before saccadic choices, representing the decision of where to move the eyes[7–10]. Furthermore, these neurons preferentially accumulate relevant information depending on the context in a cued task-switching paradigm[11]. Thus, impaired switching performance after ketamine administration might be detectable in the LIP during the context-dependent accumulation process.

We trained two macaque monkeys to switch between a direction-discrimination task and a depth-discrimination task, and investigated how neural activity of LIP neurons changed with ketamine administration while the monkey performed the cued switching task (Fig. 1). We found that the build-up for the irrelevant feature during the direction-discrimination task was more prominent after ketamine administration, in accordance with behavioral impairment. Furthermore, ketamine administration increased reaction time (RT) regardless of stimulus strength and task, and delayed the build-up onset in LIP neurons. These effects were not observed in the middle temporal (MT) area that provides sensory evidence for perceptual decisions. These results suggest that ketamine administration impairs the processing of irrelevant information in a task-specific manner, and the onset of evidence accumulation in a task-independent manner.

## Results

The monkeys performed a cued task-switching paradigm where they were instructed to discriminate either motion direction (direction-discrimination task) or stereoscopic depth (depth-discrimination task) contained in a moving random-dot stereogram depending on the color of the fixation point (Fig. 1a). Task difficulty varied by changing the percentage of dots that moved in a particular direction (motion coherence) or fell in a particular depth plane (Fig. 1b). Visual stimuli consisted of three conditions: congruent stimuli where the correct choice was the same for both tasks, incongruent stimuli where the correct choice was different depending on the task, and neutral stimuli where at least one of the two stimulus dimensions contained no information (0% motion coherence and/or 0% binocular correlation). Here, changes in motion coherence were relevant for the direction-discrimination task but were irrelevant for the depth-discrimination task. Conversely, changes in binocular correlation were relevant for the depth-discrimination task but were irrelevant for the direction-discrimination task.

**Dependence of behavioral ketamine effects on stimulus congruency.** First, we evaluated the effects of ketamine on choice accuracy and RT for congruent and incongruent stimuli. Stoet and Snyder[5] showed that ketamine reduced choice accuracy mainly for incongruent stimuli. Consistent with this observation,

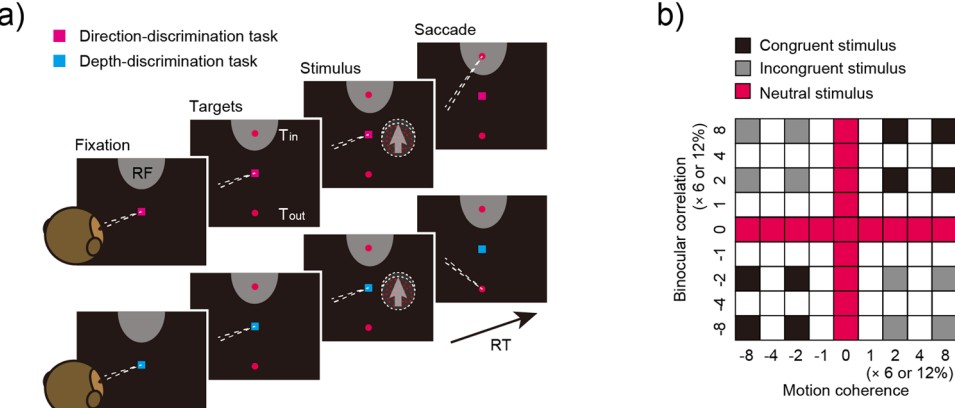

**Fig. 1 The reaction time cued task-switching paradigm. a** Reaction time cued task-switching paradigm. Monkeys performed a direction-discrimination task or depth-discrimination task according to the color of the fixation point trial by trial. When the fixation point color was magenta, the monkeys were required to report whether the dots moved upward or downward by making a saccade to the upper or lower targets, respectively (direction-discrimination task). When the fixation point color was cyan, the monkeys were required to report whether the dots were farther or nearer than the plane of fixation point by making a saccade to the upper or lower targets, respectively (depth-discrimination task). The monkeys were allowed to make a saccade at any time to report motion direction or stereoscopic depth of the visual stimulus. Tin refers to the saccade target that corresponds to either the response field of the LIP neuron or the preferred direction of the MT neuron. Tout is located diametrically opposite. RF: response field of LIP neuron, RT: reaction time. **b** Stimulus conditions used in the experiments. Filled stimulus conditions were used in this experiment. Black- and gray-colored squares denote stimulus conditions where both motion coherence and binocular correlation were non zero. The correct choice was the same for both tasks for congruent stimuli (black-colored squares), whereas the correct choice was different depending on task for incongruent stimuli (gray-colored squares). Red-colored squares denote stimulus conditions where motion coherence and/or binocular correlation was zero, and thus the stimuli were neutral. Positive motion coherences and binocular correlations refer to motion direction and depth corresponding to the Tin target, and negative motion coherences and binocular correlations to those corresponding to the Tout target.

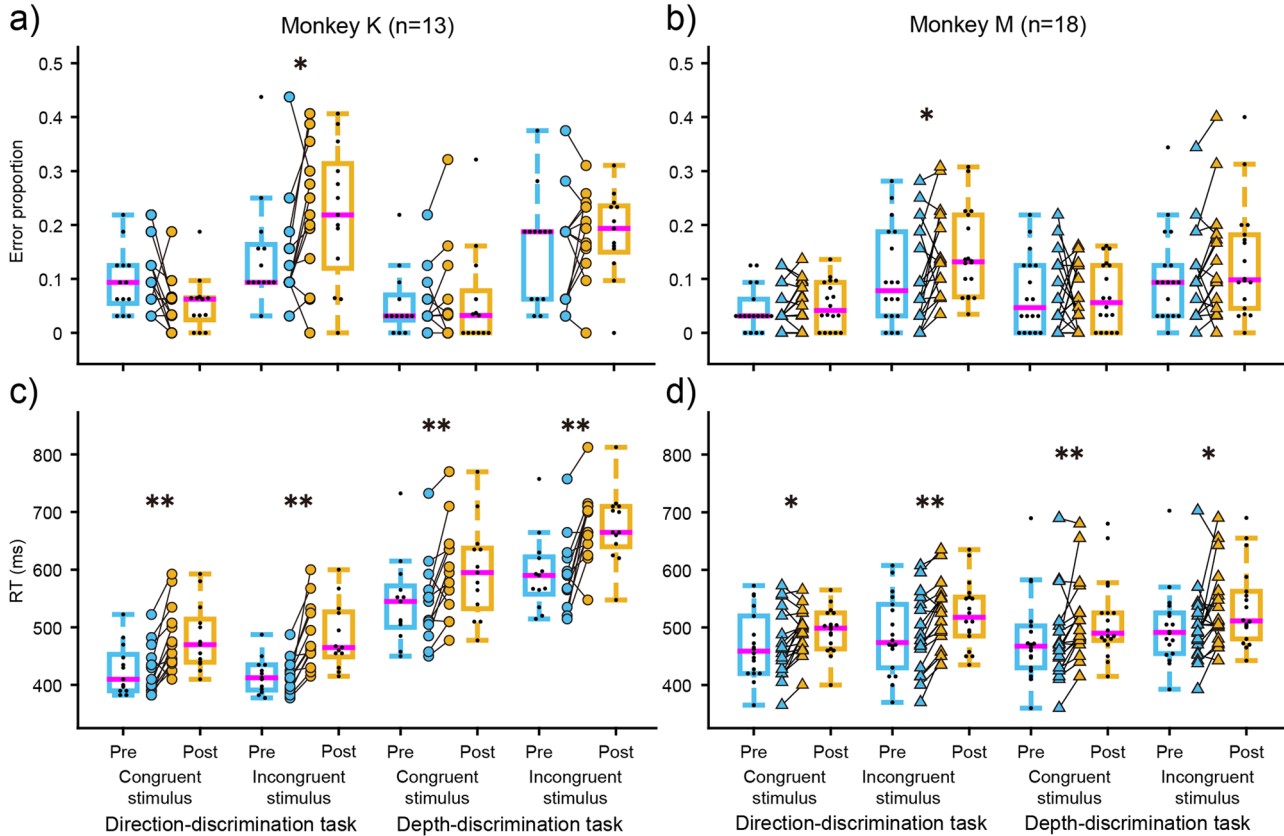

**Fig. 2 Effects of ketamine on task-switching performance depended on stimulus congruency and task.** Boxplots show error proportion (**a**, **b**) and reaction time (**c**, **d**) for congruent and incongruent stimuli with ketamine administration for the direction-discrimination task and the depth-discrimination task. The magenta lines in the box indicate median, the bottom and top edges of the box indicate lower and upper quartiles, the whiskers indicate 1.5x interquartile ranges, and the small dots indicate each experimental data. Circles and triangles denote data for monkey K (**a**, **c**) and monkey M (**b**, **d**), respectively. Asterisks indicate statistically significant results (Wilcoxon signed-rank test; \*\*$p < 0.01$, \*$p < 0.05$).

we found that, although the error proportion for congruent stimuli did not change with ketamine administration for both tasks, the error proportion for incongruent stimuli was significantly increased for the direction-discrimination task but not for the depth-discrimination task (Fig. 2a, b, Supplementary Table 1) across experiments. These modulations were not observed with saline administration (Supplementary Fig. 1). These results indicate that ketamine impaired decision making for the incongruent stimulus only during the direction-discrimination task in a task-specific manner. In addition to these task-specific effects, we also observed a task-independent effect on the RT. RT was significantly increased irrespective of task or stimulus congruency (Fig. 2c, d, Supplementary Table 1). These modulations were not observed with saline administration (Supplementary Fig. 1).

**Impaired processing of irrelevant information after ketamine administration.** Although the analysis for stimulus congruency indicated impairment of executive functioning during the direction-discrimination task, it did not address whether the impairment was due to an inability to accumulate relevant information or to a reduction in the ability to ignore irrelevant stimuli. To answer this question, we analyzed psychometric functions for neutral stimuli before and after ketamine administration during the direction-discrimination task (Fig. 3). Sensitivity to relevant stimuli was quantified by the slope of the psychometric functions using logistic regression[11,12] for visual stimuli with 0% binocular correlation (Fig. 3a, b). Sensitivity to irrelevant stimuli was quantified with visual stimuli with 0% motion coherence (Fig. 3d, e). Prior to ketamine administration,

choice was strongly dependent on relevant information and only weakly dependent on irrelevant information. However, choice was more dependent on irrelevant information after ketamine administration in both monkeys (Z test, $p < 0.001$ for monkey K, $p = 0.035$ for monkey M, Fig. 3d, e), whereas the slope of the psychometric function for relevant information was not affected (Z test, $p = 0.98$ for monkey K, $p = 0.71$ for monkey M, Fig. 3a, b). Furthermore, we tested the statistical significance of the increase in slope by comparing sensitivity (logistic regression coefficients $\beta_1$ and $\beta_2$) before and after administration in each experiment (Fig. 3g–j). The slope of the direction sensitivity for the direction-discrimination task was not changed (Wilcoxon signed-rank test, $p = 0.42$ for monkey K; $p = 1.0$ for monkey M, Fig. 3g, h), but the slope of the depth sensitivity significantly increased with ketamine administration for both monkeys (Wilcoxon signed-rank test, $p = 0.011$ for monkey K; $p = 0.031$ for monkey M, Fig. 3i, j). However, these modulations were not observed in the depth-discrimination task (Supplementary Fig. 2). Overall, only the depth sensitivity for the direction-discrimination task after ketamine administration significantly increased for both monkeys. These results demonstrate that the sensitivity to irrelevant information in the direction-discrimination task increased with ketamine administration without affecting the sensitivity of relevant information, which led to the deterioration of behavioral performance for incongruent stimulus.

**Non-decision processes are delayed after ketamine administration.** The RT analysis in Fig. 2 showed that ketamine increased

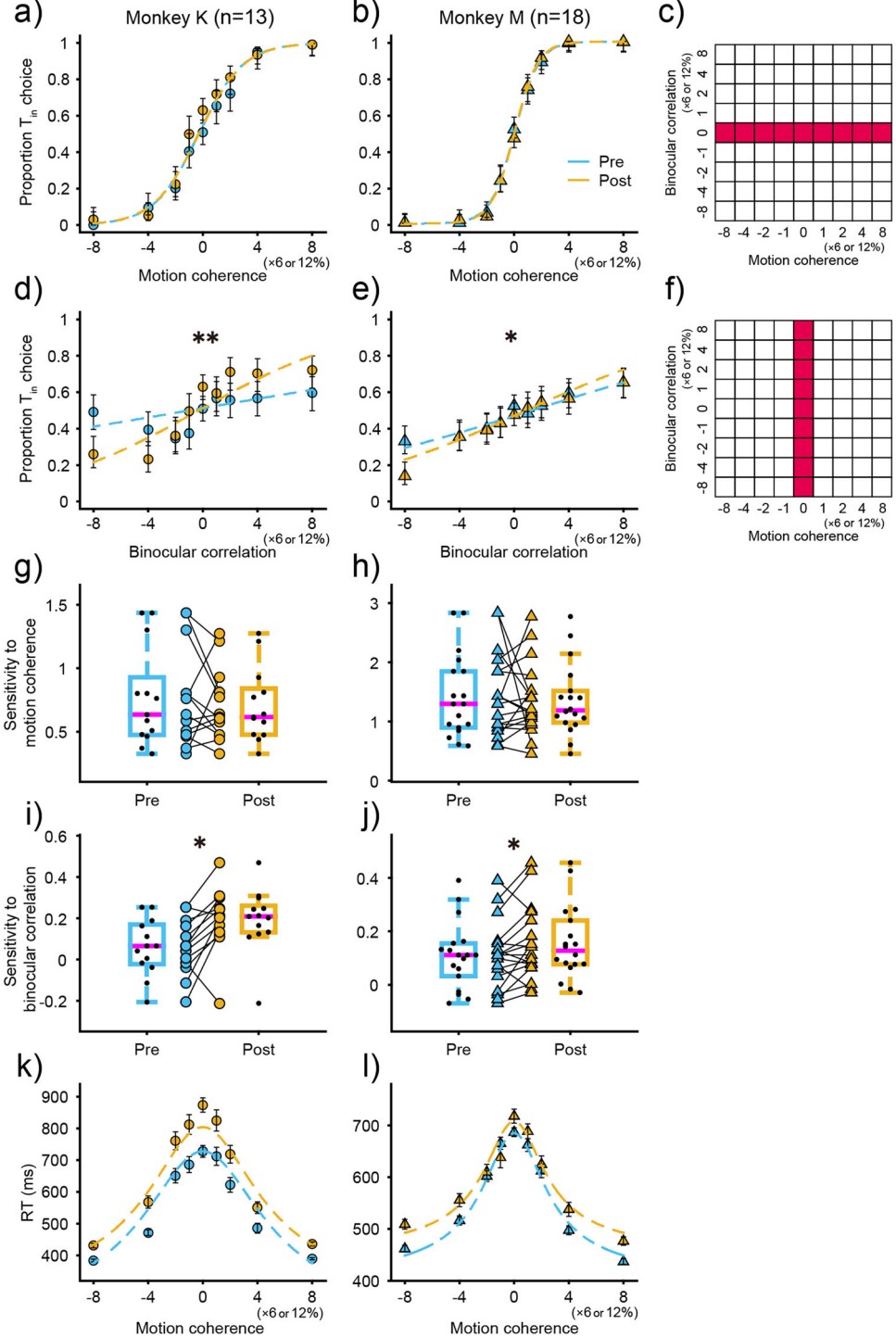

**Fig. 3 Effects of ketamine on choice and reaction time for the direction-discrimination task.** Psychometric function before (blue) and after (yellow) ketamine administration for monkey K (**a**) and monkey M (**b**). Proportion Tin choice is plotted as a function of motion coherence with 0% binocular correlation. Stimuli used to plot psychometric functions are shown as a grid (**c**). Data were combined across 13 experiments for monkey K and 18 experiments for monkey M. Lines denote logistic regression fits, and error bars indicate 95% confidence interval. **d–f** Proportion Tin choice is plotted as a function of binocular correlation with 0% motion coherence. Asterisks indicate statistically significant results (Z test; \*\*$p < 0.01$, \*$p < 0.05$). Conventions are the same as in (**a–c**). Boxplots show the effects of ketamine on sensitivity to motion coherence for the direction-discrimination task in monkey K (**g**) and monkey M (**h**). Sensitivity was derived from the slope of the psychometric function fitted for each experimental data as shown in (**a–c**). Boxplot conventions are the same as in Fig. 2. **i, j** Boxplots show the effects of ketamine on sensitivity to binocular correlation for the direction-discrimination task. The conventions are the same as in Fig. 2. Sensitivity was derived from the slope of the psychometric function fitted for each experimental data as shown in (**d–f**). Asterisks indicate statistically significant results (Wilcoxon signed-rank test; \*$p < 0.05$). Chronometric function before (blue) and after (yellow) ketamine administration for monkey K (**k**) and monkey M (**l**). RTs are plotted as a function of motion coherence. Lines denote drift diffusion model fits using all 6 parameters, and error bars indicate SEM. Conventions are the same as in (**a, b**).

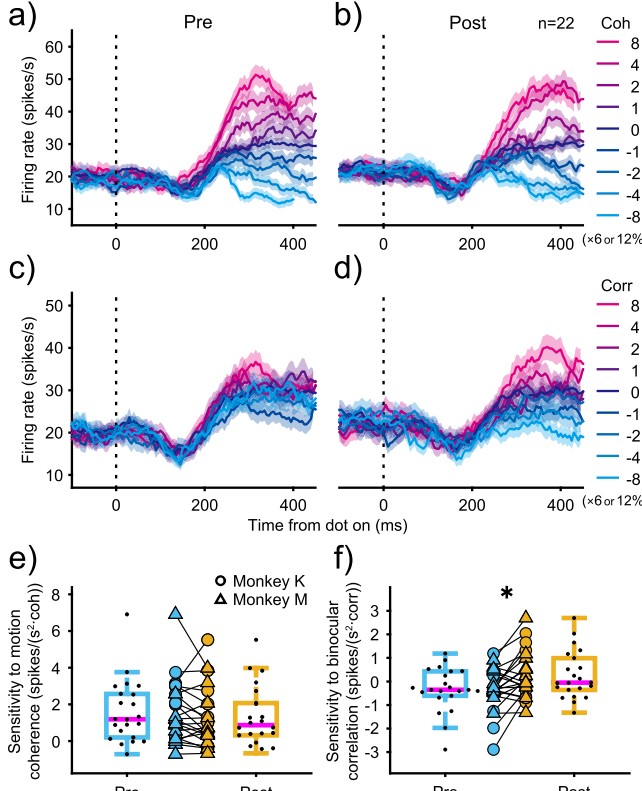

**Fig. 4 Effects of ketamine on average LIP responses for the direction-discrimination task.** Average firing rates across 22 LIP neurons aligned to visual stimulus onset are plotted before (**a**) and after (**b**) ketamine administration. Each colored line denotes firing rates at different motion coherence with binocular correlation fixed at zero. Dotted-vertical line denotes the time of visual stimulus onset. Shaded area denotes SEM. **c**, **d** Each colored line denotes firing rates at different binocular correlation with motion coherence fixed at zero. The conventions are the same as in (**a**, **b**). **e**, **f** LIP sensitivity derived from build-up slope before and after administration. Boxplots show the effects of ketamine on the sensitivity to motion coherence (**e**) and binocular correlation (**f**) for the direction-discrimination task. Boxplot conventions are the same as in Fig. 2. Circles and triangles denote each experimental data for monkey K and monkey M, respectively. Asterisks indicate statistically significant results (Wilcoxon signed-rank test; *$p < 0.05$).

RT irrespective of task or stimulus congruency. To further examine how the RT processes were impaired with ketamine administration, we focused on neutral stimuli and evaluated the dependence of RT on the strength of relevant information. Figure 3k–l shows the mean RT as a function of relevant stimulus strength for the direction-discrimination task with ketamine administration. RT significantly increased with ketamine administration for both monkeys (Wilcoxon signed-rank test, $p = 0.0002$ for monkey K, $p = 0.0084$ for monkey M), and the increment was constant regardless of stimulus strength (Fig. 3k, l). This was also observed for the depth-discrimination task (Wilcoxon signed-rank test, $p = 0.0012$ for monkey K, $p = 0.0018$ for monkey M, Supplementary Fig. 2k, l). As a consequence, the chronometric function in both tasks was shifted upward after ketamine administration. These were not observed for either task after saline administration (Wilcoxon signed-rank test, $p > 0.1$, Supplementary Fig. 3). To quantify these observations, we simultaneously fit the psychometric and chronometric functions using a simple version of the drift-diffusion model ([13], see Methods). We used six parameters: k (sensitivity for stimulus

strength), a (decision bound), and $t_R$ (non-decision time) used in Palmer et al. (2005), as well as dk, da, and $dt_R$ which describe the degree of change from k, a, and $t_R$ with drug administration. The $dt_R$ parameter best captured the effects of ketamine administration. This was confirmed by comparing the Akaike information criterion (AIC) for the full six-parameter model with five parameter models where each of dk, da, and $dt_R$ was dropped (Supplementary Table 2). The AIC increase compared to the full parameter model was largest when $dt_R$ was dropped. Most AIC for the full six-parameter model was larger than the five-parameter models after saline administration, which indicated that no additional parameter was crucial to explain the saline effects. Furthermore, the average $dt_R$ between the monkeys for the direction-discrimination task was 49.8 ms, and that for the depth-discrimination task was 50.0 ms, which was very similar to the actual RT difference observed (direction-discrimination task: 50.0 ms, depth-discrimination task: 51.2 ms). These results demonstrate that non-decision processes were delayed after ketamine administration.

**Effects of ketamine on the response of LIP neurons.** To identify the neural correlates for the behavioral effects, we analyzed decision-related responses in LIP neurons. To determine whether our LIP neuron population was representative of decision-related responses observed in previous studies, we examined the relationship between persistent activity during a delayed saccade task and decision-related activity during the switch task using a selectivity index d', and compared our population with a previous study[14]. The distribution of d' seemed to be similar to that of the previous study. We also confirmed that there was no correlation between the two d's across cells ($r^2 = 0.03$, $p = 0.27$), consistent with that study, which suggests that the neural population was similar to previous reports.

LIP neurons showed task-dependent responses consistent with a previous study[11]. Figure 4a–d shows the average responses of 22 neurons (Monkey K: 10 neurons, Monkey M: 12 neurons) aligned to the onset of the visual stimulus before and after ketamine administration during the direction-discrimination task. Before ketamine administration, the firing rates changed depending on the stimulus strength for relevant information (Z test, $p = 0.0001$). After ketamine administration, the baseline firing rate increased modestly (Pre: $19.7 \pm 16.7$ spikes/s, Post: $22.2 \pm 17.6$ spikes/s, Wilcoxon signed-rank test $p = 0.19$) but the stimulus strength-dependent change in firing rate remained (Z test, $p = 0.0056$). However, the stimulus strength-dependent change in firing rate was only observable after ketamine administration for irrelevant information (Z test, $p = 0.0022$), whereas this was not observed before ketamine administration (Z test, $p = 0.49$).

To quantify whether ketamine administration truly affected build-up activity, we compared the build-up slope before and after ketamine administration. Because build-up activity is thought to reflect both evidence accumulation and urgency, the time-dependent rise in activity independent of stimulus strength[15,16], we first subtracted the mean response across all stimulus conditions from the raw response. Using this mean-subtracted firing rate, we computed the build-up slopes by fitting lines to the responses from 200 to 400 ms after visual stimulus onset. This time window was delayed by 50 ms after ketamine administration because reaction time was delayed by 50 ms and the drift diffusion model parameter $dt_R$ was approximately 50 ms, presumably reflecting a 50 ms delay in build-up onset, which was confirmed in the analysis described in the following section. The stimulus dependence of build-up slope in the direction-discrimination task for irrelevant information was significantly larger after ketamine administration than before administration

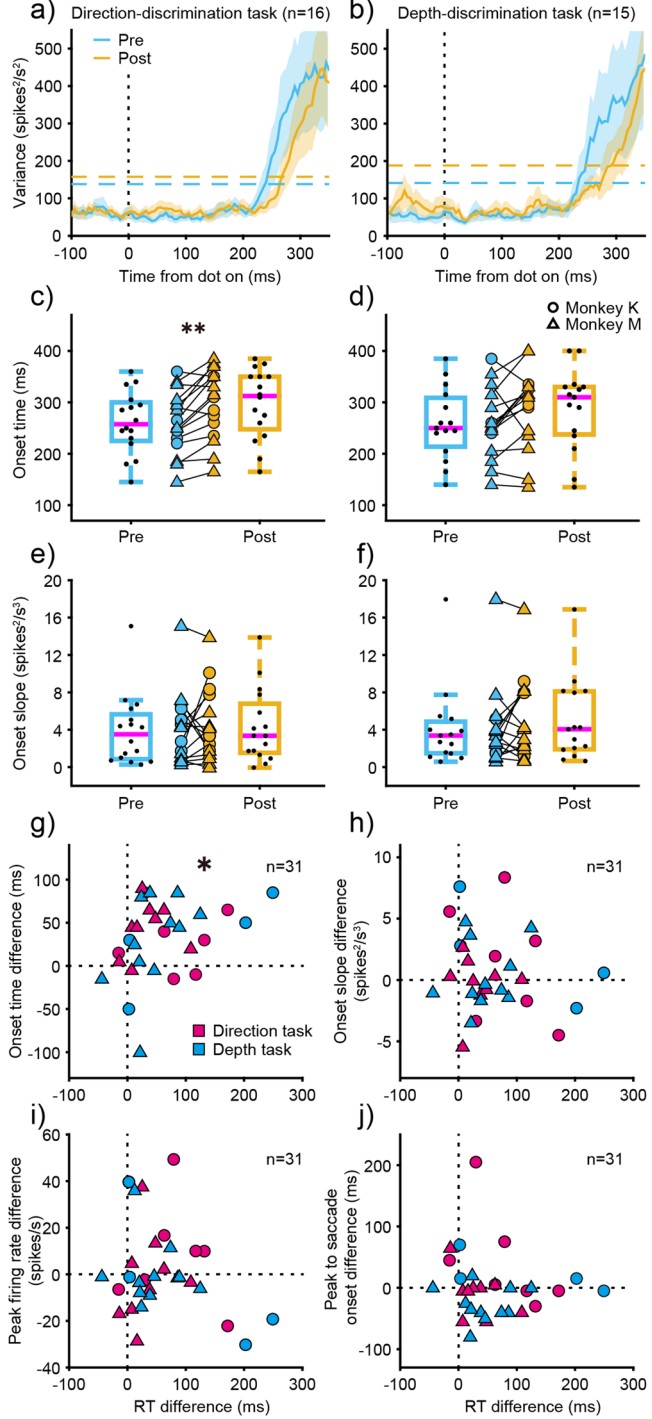

**Fig. 5 Effects of ketamine on build-up onset.** Build-up onset of LIP neurons before and after ketamine administration for the direction-discrimination task (**a**) and the depth-discrimination task (**b**). Solid lines denote firing rate variance among different motion coherences at zero binocular correlation for the direction-discrimination task, and among different binocular correlations at zero motion coherence for the depth-discrimination task aligned to visual stimulus onset. Dashed-horizontal lines denote average threshold corresponding to three standard deviations above the baseline. Blue denotes before administration, and yellow denotes after administration. Shaded area denotes SEM. Build-up onset time (**c**, **d**) and slope (**e**, **f**) before and after ketamine administration for the direction-discrimination task (**c**, **e**) and the depth-discrimination task (**d**, **f**). Onset time denotes when the variance deviated from threshold corresponding to three standard deviations above the baseline, and onset slope denotes the slope of fitted line around onset time. Circles and triangles denote each experimental data for monkey K and monkey M, respectively. Asterisks indicate statistically significant results (Wilcoxon signed-rank test; \*\*$p < 0.01$). Boxplot conventions are the same as in Fig. 2. Relationship between RT delay and build-up onset delay (**g**), build-up onset slope difference (**h**), peak firing rate difference (**i**) and peak to saccade onset difference (**j**). Magenta denotes data for the direction-discrimination task, and cyan denotes data for the depth-discrimination task. Asterisks indicate statistically significant correlation (Spearman's; \*$p < 0.05$).

**Effects of ketamine on the onset of build-up activity of LIP neurons.** As previously shown in the chronometric functions, RT was delayed by a fixed amount across stimulus strength after ketamine administration. Considering that the delay had no relationship with stimulus strength, we assumed that the delay was attributed to a non-decision process: either a delay in build-up onset or a delay in decision to saccade onset. Next, we attempted to distinguish between the two processes. To elucidate whether build-up onset was delayed with ketamine administration, we evaluated the onset of build-up activity by computing the firing rate variance among relevant stimulus strength, an index that reliably increases at build-up onset[15]. Figure 5a–b shows the time course of variance before and after ketamine administration. Build-up onset was delayed after ketamine administration for both task, whereas there was no change after saline administration (Supplementary Fig. 6). To determine when the variance deviated from baseline, we set a threshold corresponding to three standard deviations (SDs) above baseline, and defined this as onset time. Then we compared onset time before and after drug administration across experiments. Although no differences were detected after saline administration in either the direction-discrimination task or depth-discrimination task (Supplementary Fig. 6c, d), ketamine administration significantly delayed the build-up onset time for the direction-discrimination task and a similar trend was observed for the depth-discrimination task (Fig. 5c, d, Supplementary Table 3). There were no differences in the slope of the variance traces around onset time (calculated from 30 ms before to 30 ms after onset time, Fig. 5e, f), indicating that the delayed onset was not due to changes in the slope of the variance traces with ketamine. Although we used separate thresholds before and after ketamine administration, we confirmed these results when a single threshold was calculated from the baselines of both the pre- and post-conditions (Supplementary Fig. 7).

We further examined whether the build-up peak firing rate, and the time from build-up peak to saccade onset for preferred choices, changed after ketamine administration to determine whether there was a delay in decision to saccade onset. Neither peak firing rate nor the time from build-up peak to saccade onset differed after saline or ketamine administration (Supplementary

(Wilcoxon signed-rank test, $p = 0.014$, Fig. 4e), whereas no significant change was found for relevant information (Wilcoxon signed-rank test, $p = 0.095$, Fig. 4f). These results were confirmed using identical time windows (without the 50 ms delay) as shown in Supplementary Fig. 4.

No significant changes with ketamine administration were found for the stimulus dependence of build-up slope during the depth-discrimination task (Wilcoxon signed-rank test, $p = 0.73$ for relevant information; $p = 0.12$ for irrelevant information, Supplementary Fig. 5). These results indicate that ketamine administration increased sensitivity to irrelevant information in LIP neurons only during the direction-discrimination task, consistent with the behavioral results.

Table 3). Finally, we tested whether the delay in build-up onset time, the difference in onset slope, peak firing rate, or the time from build-up peak to saccade onset were related to delayed RT (Fig. 5g–j). A significant correlation was observed between the delay in RT and the delay in build-up onset (Spearman's $r = 0.39$, $p = 0.032$, Fig. 5g), whereas no correlation was detected between the delay in RT and the difference in onset slope (Spearman's $r = -0.18$, $p = 0.33$, Fig. 5h), the difference in peak firing rate (Spearman's $r = -0.05$, $p = 0.78$, Fig. 5i), and the difference in the time from build-up peak to saccade onset (Spearman's, $r = -0.18$, $p = 0.34$, Fig. 5j). These results show that the timing of build-up onset was also delayed on days that RTs were delayed with ketamine administration.

**Effects of ketamine on middle temporal neurons.** To determine whether the LIP observations could be explained by the effects of ketamine on their inputs, we examined whether neural activity of middle temporal (MT) neurons was affected by ketamine administration. Specifically, we focused on MT sensitivity to the binocular correlation and the time course of firing rate variance among motion coherence for the direction-discrimination task, because those were the parameters mainly affected in the LIP by ketamine administration. As shown in Fig. 6a–b, the average MT responses to variation in binocular correlations were similar before and after ketamine administration. It also shows the neuronal sensitivity to binocular correlation before and after ketamine administration (Fig. 6c, d). Sensitivity was evaluated by the binocular correlation-dependent increase in firing rate for each MT neuron[17]. No significant differences were observed for either the preferred or anti-preferred stimuli after saline or ketamine administration (Fig. 6c, d, Supplementary Table 4). We also measured the time course of firing rate variance before and after administration (Fig. 6e). The time when variance deviated from baseline did not change with either saline or ketamine administration (Fig. 6f). These results indicate that ketamine administration did not affect the neural responses of MT neurons.

**Effects of nystagmus on LIP neural activity.** Nystagmus occurs after ketamine administration in monkey experiments[18]. Therefore, we examined the effect of ketamine-induced changes in eye movement on LIP neural activity. As shown in Fig. 7a, the number of fixation errors did not change with saline administration, but significantly increased 3 min after ketamine administration for more than 20 min, suggesting that nystagmus did occur in our experiments. To evaluate the stability of eye movements in the trials where fixation was maintained, we calculated the distribution of intertrial variance of eye movements during the fixation period, before and after ketamine administration. The distributions were significantly different before and after ketamine administration (two-sample Kolmogorov-Smirnov test, $p < 0.0001$, Fig. 7b). The intertrial variance increased across experiments (Wilcoxon signed-rank test, $p = 0.0004$, Fig. 7c), which suggests that eye movements were affected by ketamine administration.

We evaluated the effect of eye movement on neural activity by randomly selecting trials in each data set so that the distribution of intertrial variance of eye movements was equal before and after ketamine administration (Wilcoxon signed-rank test, $p = 0.58$, Fig. 7c). Using this eye-movement equaled data set, we re-calculated behavioral sensitivity, LIP sensitivity, and LIP onset, and confirmed the increased sensitivity to irrelevant information for both behavior (Wilcoxon signed-rank test, $p = 0.008$ for monkey K; $p = 0.016$ for monkey M, Fig. 7d) and LIP slope (Wilcoxon signed-rank test, $p = 0.014$, Fig. 7d) during the

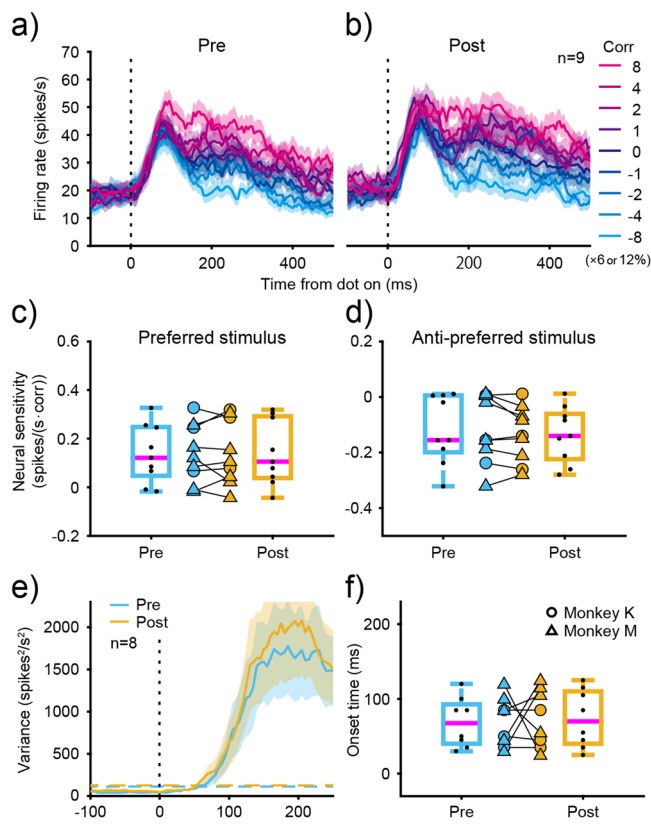

**Fig. 6 Effects of ketamine on MT responses during the direction-discrimination task.** Average firing rates of 9 MT neurons aligned to visual stimulus onset are plotted before (**a**) and after (**b**) ketamine administration. Each colored line denotes firing rates at different binocular correlation with motion coherence fixed at zero. Dotted-vertical line denotes the time of visual stimulus onset. Shaded area denotes SEM. **c**, **d** MT sensitivity to binocular correlation before and after administration. The binocular correlation-dependent increase in firing rate was evaluated for each neuron separately for preferred (**c**) and anti-preferred stimuli (**d**). Circles and triangles denote each experimental data for monkey K and monkey M, respectively. Boxplot conventions are the same as in Fig. 2. **e** Average firing rate variance of 8 MT neurons before and after administration. Firing rate variances among different motion coherence with binocular correlation fixed at zero are plotted for the direction-discrimination task. Response traces are aligned to visual stimulus onset. Dashed-horizontal lines denote threshold corresponding to three standard deviations above the baseline. Blue denotes before administration, and yellow denotes after administration. Dotted-vertical line denotes the time of visual stimulus onset. Shaded area denotes SEM. **f** Onset time of MT neurons before and after administration. Boxplots show the onset time for the direction-discrimination task before and after ketamine administration. Boxplot conventions are the same as in Fig. 2.

direction-discrimination task, as well as the delayed onset of LIP neurons (Wilcoxon signed-rank test, $p = 0.008$ for the direction-discrimination task; $p = 0.037$ for the depth-discrimination task, Fig. 7e). These results indicate that ketamine-induced changes in eye movement did not explain the increment of sensitivity to irrelevant features in LIP neurons and behavior during the direction-discrimination task, along with the delayed onset of LIP neurons.

**Asymmetry of ketamine effects.** Our results were asymmetric in that the deficits for behavioral choice were only observed in

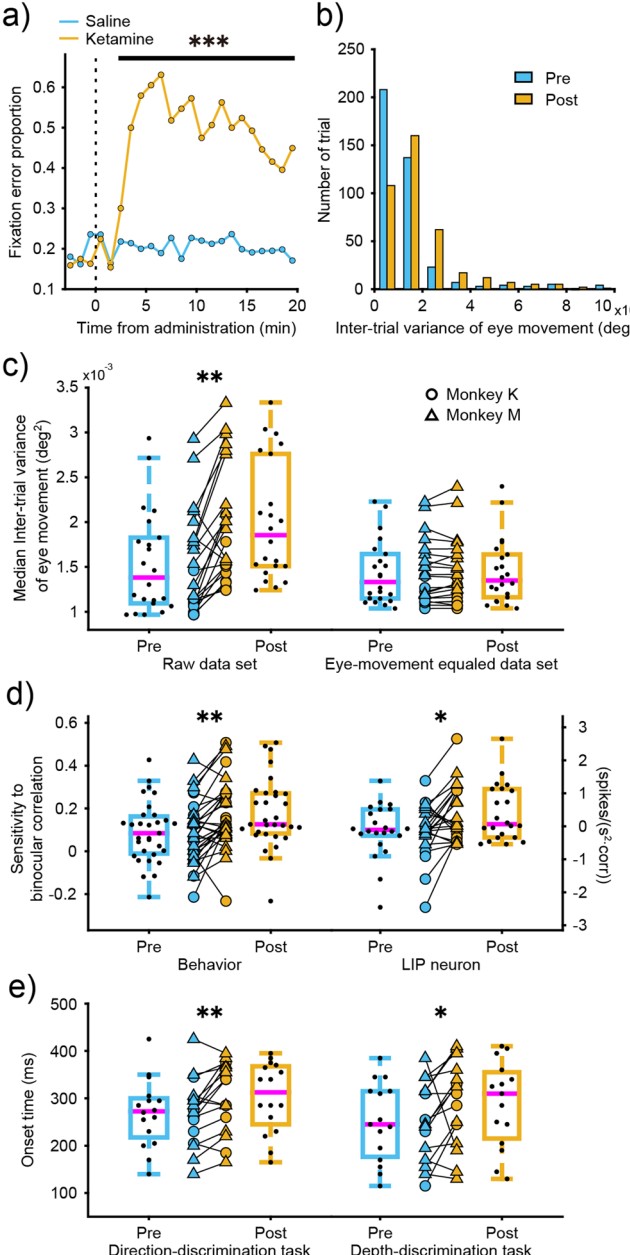

**Fig. 7 Effects of ketamine on LIP sensitivity using the eye-movement equaled data set. a** Time course of fixation error proportion during the cued task-switching task with saline (blue) and ketamine (yellow) administration. Asterisk indicates statistically significant results for ketamine administration (Chi-square test; ***$p < 0.00001$). **b** Typical histogram of intertial variance of eye movements during the fixation period with ketamine administration for an example data set. The blue bars denote the distribution of the eye movement variance before ketamine administration, and the yellow bars after ketamine administration. **c** Intertrial variance of eye movements with ketamine administration across experiments. Boxplots show the intertrial variance of eye movements during the fixation period for raw data sets and eye-movement equaled data sets. Circles and triangles denote each experimental data for monkey K and monkey M, respectively. Asterisks indicate statistically significant results (Wilcoxon signed-rank test; **$p < 0.01$). **d** Behavioral and LIP neuron sensitivity to binocular correlation during the direction discrimination task before and after ketamine administration using the eye-movement equaled data sets. Boxplots show the effects of ketamine on behavioral sensitivity and slope of LIP neuron. Circles and triangles denote the median of each equaled 1000 data sets for monkey K and monkey M, respectively. Asterisks indicate statistically significant results (Wilcoxon signed-rank test; **$p < 0.01$, *$p < 0.05$). Boxplot conventions are the same as in Fig. 2. **e** Onset of LIP neuron before and after ketamine administration using the eye-movement equaled data sets. Boxplots show the effects of ketamine on LIP onset for the direction-discrimination task and the depth-discrimination task.

depth-discrimination task. This became more evident when we compared the time-course of variance (Supplementary Fig. 9c). The variance for the depth-discrimination task increased at about 200 ms and peaked at 300 ms after visual stimulus onset, whereas the increase in variance for the direction-discrimination task was gradual at about 200 ms after visual stimulus onset and peaked at about 600 ms.

We also plotted the time-course of variance during the direction-discrimination task before and after ketamine administration (Supplementary Fig. 9d). Although the variance pre-administration gradually increased in a similar way, variance post administration quickly increased after ketamine administration. These results suggest that the monkeys were successful in suppressing irrelevant depth information for a few hundreds of milliseconds during direction discrimination, but the suppression was impaired with ketamine administration. Overall, the task-dependent difference in the way irrelevant information was accumulated may have had an effect on the asymmetric deficits for behavioral choice.

## Discussion

We demonstrated pronounced effects of ketamine administration on the build-up activity in the LIP; specifically, we found increased sensitivity to irrelevant information and delayed onset of evidence accumulation after ketamine administration. The consequences of both the increased sensitivity to irrelevant information and delayed onset of evidence accumulation were evident in behavior, which suggests that the behavioral modulation could be attributed to changes in LIP build-up activity. It is noteworthy that the sensitivity and onset of MT neuron activity were not affected by ketamine administration. Although this was not surprising given that MT responses do not change depending on the task[12], our results suggest that ketamine acts on neural processes beyond sensory representation in the MT area, presumably directly on neural processing of decision formation.

the direction-discrimination task. To elucidate whether the asymmetric effects of ketamine could be explained by the difference between the two tasks, we examined the behavioral performance between repetition and switch trials (Supplementary Fig. 8). Although we compared the error proportion and reaction time between the repetition and switch trials in each monkey, there were no differences between them (Supplementary Table 5).

We further elucidated the difference in LIP accumulation between the two tasks. There seemed to be a task-dependent difference in the way LIP neurons accumulated irrelevant information. We compared the LIP responses when binocular correlation changed for the direction-discrimination task (Supplementary Fig. 9a), and when motion coherence changed for the depth-discrimination task (Supplementary Fig. 9b). LIP neurons started to respond at about 200 ms after visual stimulus onset. However, strength-dependent build-up activity was weak for the direction-discrimination task, whereas this was evident during the

Importantly, our results suggest that a control mechanism exists for the onset of evidence accumulation. This type of control would be useful in cases where time is needed to correctly determine what information is relevant to prepare for adequate accumulation. Our results suggest that ketamine can interfere with evidence accumulation onset or resetting of the integrator that accumulates evidence that may be important for temporal regulation of evidence accumulation.

Our results were asymmetric in that increased sensitivity to irrelevant information was only observed in the direction-discrimination task. This may be due to asymmetries in switching between tasks. Although we attempted to equate the difficulty of the two tasks, asymmetries may occur when switching between tasks; that is, it is easier to switch from one task to the other than vice versa[19–21]. Asymmetries can be found in switch costs[19,20]; however, we did not observe them in our study. This is presumably because the intertrial interval was too long to observe the switch cost. If the intertrial interval was very short, we may observe asymmetric switch costs as previously reported[3,21].

We further investigated asymmetries in LIP responses and found a task-dependent difference in the way irrelevant information was accumulated. This asymmetry might be attributed to the order of task training. We trained the monkeys to discriminate motion direction first, and subsequently trained them to discriminate stereoscopic depth. Whether the order of training actually contributes to the way irrelevant information is accumulated should be investigated in future research.

Stoet and Snyder reported that ketamine affects accuracy and RT for stimuli with conflicting behavioral goals (incongruent stimuli)[5]. Our study is consistent with this result, and importantly, provides a neural account. Specifically, ketamine affected build-up activity of LIP neurons to irrelevant information in a way that would improve accuracy and shorten RT for congruent stimuli, whereas it impaired accuracy and lengthened RT for incongruent stimuli. In reality, RT was elongated for congruent as well as incongruent stimuli, but this could be explained by considering the delay in build-up onset.

It is difficult to precisely address where in the brain ketamine had its main effect. Previous imaging studies using positron emission tomography and functional magnetic resonance imaging (MRI) showed that acute ketamine administration activates frontal and cingulate cortices in healthy participants[22,23]. Furthermore, the changes were prevented with a glutamate release inhibitor, which suggests that excessive glutamate release by ketamine administration is involved in abnormal activation[24]. The prefrontal cortex (PFC) represents abstract task rules in individual neurons[25,26] and synchrony of neural oscillations[21], which indicates that the functions of the frontal and anterior cingulate cortices are central to flexible decision making[27,28]. Presumably, ketamine affected neurons in those areas that regulate LIP build-up activity. Indeed, systemic ketamine injection not only reduces the task selectivity of PFC neurons[29] but also affects lower band oscillations important for long-range communications in monkeys during an anti-saccade task[30,31]. Nevertheless, ketamine may have had direct effects on LIP responses in addition to effects on other cortical areas. It is also difficult to address through which receptor ketamine had its main effect. Although ketamine is mainly a NMDA receptor antagonist, it also affects other receptors such as α-amino-3-hydroxy-5-methyl-4-isoxazole propionic acid (AMPA) receptors[32].

The neural circuits underlying increased sensitivity to irrelevant information, and delayed onset of evidence accumulation after ketamine administration are of considerable interest. Wang provided a decision-making model that incorporates NMDA receptors[33]. Blockage of NMDA receptors based on this model predicts decreased sensitivity to relevant information and a decrease in the slope of build-up activity. Here, we only found impairments for irrelevant information, but this could depend upon ketamine dosage: at a higher dose, the sensitivity to relevant information may also be affected. Increased sensitivity to irrelevant information may be explained by assuming mutual inhibition between sets of integrators for each task. Weakening inhibition might unbind sensitivity to irrelevant information. It is unclear how inhibiting NMDA receptors might affect evidence accumulation onset in this model, which should be investigated in future research.

Finally, administration of low-dose ketamine induces schizophrenia-like behavior in human participants[34]. Although it is unclear how ketamine-administered humans behave in a task-switching paradigm, our results are similar to studies of patients with schizophrenia in that they show increased difficulty responding to incongruent stimuli[4,6,35]. Therefore, the deficits that we found in this study may be directly related to cognitive deficits observed in patients with schizophrenia. Specifically, increased sensitivity to irrelevant information, and delayed onset of evidence accumulation may be major factors related to schizophrenia. This hypothesis warrants investigation in future studies.

## Methods

**Subjects and surgery**. Two male Japanese macaques (*Macaca fuscata*), weighing 6 kg (monkey K) and 7 kg (monkey M), were used in this study. After training the monkeys to sit calmly in a dedicated chair, they were prepared for experiments using standard surgical procedures. We attached a head post to restrain head movement, and implanted a coil wire under the left conjunctiva to monitor eye movements[36]. To access the LIP and the MT, we mounted a recording chamber at an angle of 25° above the horizontal, and positioned it over the occipital cortex around ~12 mm lateral and 21 mm anterior to the occipital ridge for the LIP, and ~17 mm lateral and 14 mm dorsal for the MT[11,37]. We also used structural MRI to confirm placement of the chamber. All of the animal care and experimental procedures were approved by the Juntendo University, Tamagawa University and University of Yamanashi Animal Care and Use Committee, and were in accordance with the National Institutes of Health guidelines.

**Visual stimuli**. The monkeys faced a 22-inch cathode ray tube monitor (HM204DA; Iiyama), which was placed 57 cm away from their eyes with a visual angle of 40° × 30°. Visual stimuli of random-dot stereograms were generated by an OpenGL accelerator board with quad-buffered stereo support (Quadro FX 1400; NVIDIA). Dot density was 64 dots per square degree per second, and each dot subtending ~0.1° above the horizontal, and positioned it over the occipital cortex view was achieved by alternately viewing the stereo half-images for the left and right eyes at a refresh rate of 100 Hz through a pair of ferroelectric liquid crystal shutters (Micron) that were synchronized to the video input. To enable precise binocular disparities and smooth movements, we plotted dots with sub-pixel resolutions using anti-aliasing provided by the OpenGL board. We also used red dots on a black background to minimize ghosting effects.

**Behavioral tasks and training**. We used a commercially available software package (TEMPO; Reflective Computing) for controlling behavioral tasks and data acquisition, and MATLAB (MathWorks) for the online data analyses. Eye position was monitored using a magnetic search coil system (Sankeikizai) and stored at 200 Hz.

After fixation training, we trained the monkeys to individually discriminate between motion direction (direction-discrimination task) and stereoscopic depth (depth-discrimination task). During the direction-discrimination task, the monkeys were required to report whether the motion of moving dots was upward or downward by making a saccade to the upper or lower targets, respectively. During the depth-discrimination task, the monkeys were required to report whether the stereoscopic depth of dots was farther or nearer than the plane of fixation point by making a saccade to the upper or lower targets, respectively. The speed of coherently moving dots was 8°/s, and the binocular disparity was either −0.5° or 0.5°.

After the training for each task was completed, we started the training of switching between the two tasks, which was instructed by the color of the fixation point; magenta indicated the direction-discrimination task and cyan indicated the depth-discrimination task. Figure 1a shows an example. The $T_{in}$ target was positioned in the response field (RF) of the LIP neuron or towards the preferred direction of the MT neuron at 10 degrees eccentricity, whereas the $T_{out}$ target was positioned diametrically opposite to the $T_{in}$ target. In this example, the dot stimulus moved upward, the stereoscopic depth was nearer than the fixation point, and $T_{in}$

was located in the RF of the LIP neuron which was above the fixation point. When the color of the fixation point was magenta, i.e., direction-discrimination task the monkey was required to report the motion direction; thus, the correct choice was the upward target ($T_{in}$). Conversely, when the color of the fixation point was cyan, i.e., depth-discrimination task, the monkey was required to report the stereoscopic depth; thus, the correct choice was the downward target ($T_{out}$). The monkey had to make a different choice according to the color of the fixation point, even though the presented stimulus was the same. We varied the task difficulty by manipulating the coherence of moving dots (motion coherence) and the binocularly correlated dots (binocular correlation) in the stimulus to examine the parametric effects of the stimulus (Fig. 1b). Motion coherence and binocular correlation were selected according to the table in Fig. 1b where positive values indicated coherence/correlation with motion or depth corresponding to the $T_{in}$ choice, whereas negative values indicated coherence/correlation with motion or depth corresponding to the $T_{out}$ choice. For example, the top right black square in Fig. 1b denotes that the stimulus had a motion coherence of 48% or 96% ($8 \times 6\%$ or $8 \times 12\%$) and a binocular correlation of 48% or 96% ($8 \times 6\%$ or $8 \times 12\%$), and the bottom right gray square denotes that the stimulus had a motion coherence of 48% or 96% ($8 \times 6\%$ or $8 \times 12\%$) and a binocular correlation of −48% or −96% ($-8 \times 6\%$ or $-8 \times 12\%$). In the example shown in Fig. 1a, positive motion coherences correspond to upward motion and negative motion coherences correspond to downward motion. Likewise, positive binocular correlations correspond to the far depth and negative binocular correlations correspond to the near depth. Therefore, the squares in the upper right and lower left (black-colored stimuli) correspond to stimuli where the correct choice was the same for both tasks. Thus, they are referred to as congruent stimuli. Conversely, the squares in the upper left and lower right (gray-colored stimuli) correspond to stimuli where monkeys had to make different choices depending on task. Thus, they are referred to as incongruent stimuli. For the red-colored stimulus conditions, stimuli were neutral in that they contained either motion or depth information (i.e., the stimulus strength for motion direction and/or depth was zero). Neutral (red-colored) conditions contained 36 trials, and congruent (black-colored) and incongruent (gray-colored) conditions contained 8 trials each in a block. Therefore, a total of 52 trials were used for each of the two tasks, yielding 104 trials within a block. These stimulus conditions were pseudorandomly interleaved within one block.

The monkeys had to quickly switch between tasks to perform correctly because the color cue was randomly changed from trial to trial. Each trial started with presentation of the fixation point. The two choice targets appeared when the monkey fixated the point for 300 ms. After a random delay (0.3–2.8 s), which was drawn from a truncated exponential distribution, the random-dot stimulus was presented within a circular aperture (8° in diameter) positioned 8° from the fixation point for LIP experiments. The stimulus position was to the left of the fixation point for monkey M and to the right for monkey K. The random-dot stimulus was presented within the receptive field for MT experiments. Once the stimulus was presented, the monkey was able to choose the target with a saccade at any time, and a correct choice was rewarded by a drop of water or juice. The trial was aborted if their gaze left the window of fixation (1° from the fixation point).

We also trained the monkeys for memory-guided saccade task, which was used to identify the response field (RF) of individual LIP neurons. While the monkeys fixated, a target was presented for 300 ms at one of eight positions that was evenly distributed 12° around the fixation point. The monkey had to remember the position of the target for 1 s, and was rewarded with a drop of water or juice if they made a saccade to the correct target position when the fixation point disappeared.

### Electrophysiological recordings.
We recorded extracellular activity from isolated neurons in area LIP and MT using a tungsten microelectrode (FHC) with impedance values of 0.5–1.0 MΩ. We inserted the electrode forward into the cortex through a transdural guide tube using a pulse motor micromanipulator (PC-5N; Narishige), which was placed on the recording chamber. After the raw signals were amplified, single neurons were isolated online from waveforms sampled at 40 kHz using an online spike sorting software (OmniPlex Software, PLEXON Inc.). Using the memory-guided saccade task, we functionally identified the ventral portion of area LIP, and focused on neurons that responded to visual stimulus, during the delay or just before saccade. The MT area was also functionally identified from the physiological response (direction, speed, horizontal disparity, and receptive field properties)[38].

### Experimental protocols.
We first isolated LIP neurons while the monkey performed the memory-guided saccade task. To identify the RF of the neuron and examine whether the neuron exhibited delay activity for the saccade to the RF, we monitored the peristimulus time histogram for the saccade to the target in eight directions and constructed a tuning curve during the delay period online. Then we recorded neural activity during the cued task-switching paradigm before ketamine or saline administration. According to the position of the RF, we set the $T_{in}$ target in the RF of the neuron and the $T_{out}$ target in the diametrically opposite position of the $T_{in}$ target. For the isolated MT neurons, we quantitatively measured direction tuning, speed tuning, horizontal disparity tuning, receptive field location, and size tuning (area summation). We matched the visual stimulus properties to the preference of the neuron.

To parametrically manipulate the strength of the stimulus features, we varied the motion coherence among 0%, 6%, 12%, 24%, and 48% for 38 neurons; and among 0%, 12%, 24%, 48%, and 96% for 29 neurons with two directions (up or down). The binocular correlation was varied among 0%, 6%, 12%, 24%, and 48% for 32 neurons; and among 0%, 12%, 24%, 48%, and 96% for 35 neurons with two disparities (far or near).

We collected data for four blocks, which lasted about 30 min, before ketamine or saline administration. Then we administered saline or ketamine intramuscularly into their right biceps femoris muscle. The ketamine dosages were 0.25–0.5 mg/kg (Monkey K: 0.25 or 0.5 mg/kg, Monkey M: 0.35, 0.4 or 0.5 mg/kg) in each experiment, and ketamine administration was separated by at least 2 days to avoid cumulative dosing effects[39]. After administration, we recorded up to four blocks. Saline and ketamine were administered in a random order. The volume for saline and ketamine was identical. Both monkeys performed the switching task continuously after ketamine administration across all dosages, although fixation errors caused by nystagmus increased after administration (Fig. 7). Fixation error significantly increased after 3 min of ketamine administration (Chi-square test, $p < 0.00001$). Therefore, we excluded trials performed in the first 2 min after administration. This procedure excluded an average of 21 trials (21 ± 2.7 [mean ± SD]), that is 5% of the four blocks of trials post-administration.

### Statistics and reproducibility.
We recorded extracellular activity from 44 isolated neurons in the LIP area with 22 saline administration (Monkey K: 10 neurons, Monkey M: 12 neurons) and 22 ketamine administration (Monkey K: 10 neurons, Monkey M: 12 neurons) experiments. We also recorded from 23 isolated neurons in the MT area with 14 saline administration (Monkey K: 5 neurons, Monkey M: 9 neurons) and 9 ketamine administration (Monkey K: 3 neurons, Monkey M: 6 neurons) experiments. We used 36 saline and 31 ketamine administration experiments covering both LIP and MT recordings for behavioral analysis. In order to statistically test whether there were ketamine effects before and after administration, we used two-sided non-parametric statistical tests: Wilcoxon signed-rank test, Wilcoxon rank sum test, or two-sample Kolmogorov-Smirnov test. All analysis were performed with MATLAB 2014a (MathWorks).

### Behavioral data analysis.
To examine the dependence of the ketamine effect on stimulus congruency, we quantified error rate and RT for congruent and incongruent stimulus separately. Furthermore, we quantified behavioral performance with a psychometric and chronometric function, each of which demonstrated a relationship among signal strength, choice, and RT. For the psychometric function, we used logistic regression to determine the change in choice before and after saline or ketamine administration using neutral stimuli (Fig. 1b) as follows:

$$P_{stim} = \frac{1}{1 + e^{-Q}} \tag{1}$$

$Q = \beta_1 |Coh|$ (evaluation of the effect of motion coherence on choice at %binocular correlation) (2)

$Q = \beta_2 |Corr|$ (evaluation of the effect of binocular correlation on choice at 0% motion coherence), (3)

where Coh denotes motion coherence and Corr denotes binocular correlation. Logistic regression functions were fit separately for each task (direction-discrimination task and depth-discrimination task), before and after ketamine or saline administration. We used $\beta_1$ and $\beta_2$ as indices for behavioral sensitivity to each stimulus feature.

RT was measured as the time from visual stimulus onset to saccade beginning when the gaze left the fixation window. Based on a simple version of the drift-diffusion model[13], we fit the psychometric and chronometric function on trials where the strength of the irrelevant stimulus was 0% (Fig. 1b) as follows:

$$P_{stim}(x) = \frac{1}{1 + e^{-2A \cdot Kx}} \text{(Psychometric function)} \tag{4}$$

$$t_T(x) = \frac{A}{Kx} \tanh(A \cdot Kx) + T_R \text{(Chronometric function)} \tag{5}$$

and, A, $T_R$, and Kx were defined as follows:

$$A = a + da \cdot Drug \tag{6}$$

$$T_R = t_R + dt_R \cdot Drug \tag{7}$$

$$Kx = k \cdot |Coh| + dk \cdot |Coh| \cdot Drug \text{(Direction discrimination task)} \tag{8}$$

$$Kx = k \cdot |Corr| + dk \cdot |Corr| \cdot Drug \text{(Depth discrimination task)}, \tag{9}$$

where a denotes decision bound, $t_R$ denotes non-decision time, k denotes sensitivity for relevant stimulus strength, and da, dk, and $dt_R$ denote the degree of change from a, k, and $t_R$ with drug administration. We iteratively adjusted these six parameters to maximize log likelihood $\ln(L)$, which was the sum of log likelihood of the psychometric function $L_P$ and the chronometric function $L_T$ calculated as

follows:

$$L_p(x) = \frac{n!}{r!(n-r)!} P_{stim}(x)^r (1 - P_{stim}(x))^{n-r} \qquad (10)$$

$$L_T(x) = \frac{1}{\sigma_t \sqrt{2\pi}} e^{-[t_T(x) - \bar{t}_T(x)]^2 / 2\sigma_t^2} \qquad (11)$$

$$\ln(L) = \sum_x \ln\left[L_p(x)\right] + \ln\left[L_T(x)\right], \qquad (12)$$

where n denotes the number of trials, r denotes the number of correct trials, and $\bar{t}_T(x)$ denotes mean RT. The predicted standard error of the mean $\sigma_t$ was calculated as $\sigma_t = \sqrt{VAR\left[T_T(x)\right]/n}$, where $T_T$ denotes the variability in the RT. We assumed that $T_T$ was composed of the variability of the decision time $T_D$ and residual non-decision time $T_R$ as follows:

$$VAR(T_T) = VAR(T_D) + VAR(T_R) \qquad (13)$$

$$VAR(T_T) = \frac{[A \cdot \tanh(A \cdot \mu) - A \cdot \mu \cdot sech(A \cdot \mu)^2]}{\mu^3} + VAR(T_R)\,(for\,\mu{\neq}0) \quad (14)$$

$$\lim_{A \cdot Kx \to 0} VAR(T_T) = \frac{2}{3} A^4 + VAR(T_R)\,(for\,\mu = 0), \qquad (15)$$

where $\mu$ denotes drift rate, which was calculated as $\mu = A \cdot Kx$. To quantify which of the three parameters, dk, da, or $dt_R$, mainly explained modulation by ketamine, we compared relative goodness of fit of models when each of them was excluded using the AIC[40]. We calculated AIC as follows:

$$AIC = -2\ln L + 2k, \qquad (16)$$

where L denotes the maximum likelihood and k denotes the number of parameters.

**Neuronal data analysis**. We evaluated neural activity with the firing rates from single unit activity. The firing rates were calculated within a 50 ms window stepping by 5 ms, aligned to the visual stimulus onset and beginning of the saccade. We estimated the build-up slope using linear regression using the response from 200 ms to 400 ms after visual stimulus onset to estimate the build-up slope. Because the RT was delayed by approximately 50 ms and the drift diffusion model parameter $dt_R$ was approximately 50 ms with ketamine administration, this time window was delayed by 50 ms after ketamine administration for both tasks assuming that build-up onset was delayed by 50 ms. Furthermore, to estimate whether the sensitivity to motion coherence and binocular correlation changed with saline or ketamine administration, the build-up slopes for neutral stimuli were fit using linear regression as follows:

$$Buildup\ slope_{motion\ coherence} = \alpha_0 + \alpha_1 Motion\ coherence \qquad (17)$$

$$Buildup\ slope_{binocular\ correlation} = \alpha_2 + \alpha_3 Binocular\ correlation, \qquad (18)$$

where $\alpha_{0-3}$ are fitted parameters. Logistic regression functions were fit separately for each task (direction-discrimination task and depth-discrimination task), before and after ketamine or saline administration.

We quantified the onset of build-up activity by computing the variance of the firing rate among responses to different motion coherences for the direction-discrimination task and different binocular correlations for the depth-discrimination task as previously reported[15]. To determine when the variance deviated from baseline, we set a threshold corresponding to three SDs above baseline, defined as the average variance from 200 ms before to visual stimulus onset. There were some data samples in which we could not determine the onset time because the variance did not deviate from baseline. Thus, for onset analysis with saline administration, we excluded 7 and 8 data samples for the direction-discrimination task and depth-discrimination task, respectively, and with ketamine administration, we excluded 6 and 7 data samples, respectively. With these data sets, we examined the build-up peak firing rate, and the time from build-up peak to saccade onset using the response for preferred choices from 200 ms before to the saccade time. All analyses were performed using two-sided non-parametric statistical tests (Wilcoxon signed-rank test).

We quantified neural sensitivity to the binocular correlation of MT neurons as previously reported[17]. The binocular correlation-dependent increase in firing rate was calculated from 50 ms after visual stimulus onset to saccade onset using the data from stimulus conditions where the motion coherence fixed at 0 (Fig. 1b). The firing rates for preferred and anti-preferred stimuli were fit using linear regression as follows:

$$Sensitivity\ to\ binocular\ correlation_{pref} = \gamma_0 + \gamma_1 Pref\ stimulus\ strength \qquad (19)$$

$$Sensitivity\ to\ binocular\ correlation_{anti-pref} = \gamma_2 + \gamma_3 Anti-pref\ stimulus\ strength, \quad (20)$$

where $\gamma_{0-3}$ are fitted parameters. Logistic regression functions were fit for the direction-discrimination task before and after ketamine or saline administration. We used $\gamma_1$ and $\gamma_3$ as indices for sensitivity to binocular correlation for preferred and anti-preferred stimuli, respectively.

**Eye movement data analysis**. To examine the effect of ketamine on eye movement, we evaluated the number of fixation error trials and the stability of eye movements within a trial. Fixation error trials were defined as trials where the monkey's gaze left the fixation window (1° from the fixation point) before the visual stimulus onset. The stability of eye movements was quantified using the intertrial variance of eye movements calculated as the sum of the horizontal and vertical variance from 0 ms to 300 ms after visual stimulus onset. We used two-sample Kolmogorov-Smirnov test for statistical comparison between the distribution of the intertrial variance before and after ketamine administration. To equalize the degree of eye movements with ketamine administration, we generated an eye-movement equaled data set by randomly selecting trials so that the distribution of intertrial eye movement variance was equal before and after administration in each experiment. After sorting the pre- and post-administration intertrial eye movement variance distributions into equally spaced bins, we randomly collected an equal number of trials for each bin from both distributions, and re-calculated the behavioral sensitivity, build-up slope, and onset of LIP neurons. We performed this process 1000 times, and compared the median behavioral sensitivity, build-up slope, and onset of LIP neurons before and after administration.

**Reporting summary**. Further information on research design is available in the Nature Research Reporting Summary linked to this article.

## Data availability

The source data underlying the graphs in the main figures are available in Supplementary Data 1–6. All other data are available from the corresponding author upon request.

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

## Acknowledgements

This study was supported, in part, by the JST PRESTO and JSPS KAKENHI (Grant Numbers: JP15H01447, JP26290008, JP19H05207, JP19H03531 and JP21K15608). We thank Haruyo Kimizuka for her technical and surgical assistance, and Hironori Kumano for his comments on the manuscript.

## Author contributions

Y.S. and T.U. designed experiments, Y.S. collected and analyzed data, Y.S. drafted this work, T.U. critically revised this work, Y.S. and T.U. approve the final version.

## Competing interests

The authors declare no competing interests.
