## [Peer Review File · Communications Biology]

Reviewers' comments:

Reviewer #1 (Remarks to the Author):

The authors of The NMDA receptor antagonist ketamine impairs and delays context-dependent decision making in the parietal cortex studied how systemic administration of subanesthetic Ketamine affected behavioral and neural process of decision making while primate switched between direction-discrimination task and depth-discrimination task. Interestingly, they found that Ketamine administration didn't change monkey's sensitivity to relevant stimuli, but changed their sensitivity to irrelevant stimuli on behavioral level. In accordance with behavior, Ketamine administration increased LIP neural sensitivity to irrelevant stimuli. They also found that delayed reaction time after ketamine injection may be due to delayed onset of decision related neural activity. Their MT data didn't find similar results as what found in LIP data, excluding the possibility that their finding was due to ketamine effects on LIP's input region. They also generated the eye-movement equaled dataset to exclude the possible explanation of their finding by nystagmus effects. Their findings are interesting and important to our understanding about how ketamine modulate neural activity and decision making in primate brain. I don't have major concerns, just some minor suggestions listed below:

1. The abstract part looks too short, and didn't include their main findings point by point.
2. The authors didn't write how long the primate waited between ketamine injection and starting recording. Ketamine need some minutes to get into brain.
3. The task description is not clear enough to me. Can the authors provide detailed stimuli pattern respective to every black and gray squares in Fig. 1b?

Reviewer #2 (Remarks to the Author):

It has been previously shown that low dose ketamine impairs performance on difficult tasks. In particular, it increases interference between tasks. This study investigates the neural locus of this effect. Correlates of increased interference from depth cues in a motion-discrimination task are identified in LIP and are absent in MT. The paper is well written and in the text, the results are clear and unambiguous. There are significant issues in two of the figures, but these will be easily fixed. The description of the analyses can be improved (again, easily done). Some issues specific to switch paradigms remain, and might require a bit more analysis. Even as currently written, however, the findings will advance work aimed at identifying how primates use contextual cues to select out relevant stimuli and ignore irrelevant stimuli, with substantial ramifications relevant to many different fields and research programs.

1. Definitions:

Congruent and incongruent stimuli are not defined. Similarly, relevant and irrelevant stimuli are not defined. While the general terms are clear, readers who have not previously encountered dual tasks will be confused. It would help to explicitly state that in both tasks, stimuli have both a depth and a motion, but that depth is irrelevant in the motion task and motion is irrelevant in the depth task.

2. Figure 3: This critical figure is very unclear.

a) In panel a, the blue dashed lines are not visible.

b) Why is choice accuracy for the relevant stimulus identical pre- and post-ketamine? I thought the message is that, post-ketamine, there is interference from the irrelevant stimulus that degrades performance? Why don't we see this in either animal?

c) I was very confused by how "choice bias directed by the irrelevant stimulus feature" could be plotted in terms of choice accuracy. Now I think I see that perhaps there are two different Y axes, one for the dashed lines (accuracy) and one for the solid lines (bias). However, this still does not explain what is actually being plotted. Explicit definitions for "accuracy" and "bias" would be very helpful.

d) Panel b plots sensitivity, which we are told is "derived from the slope ... in (a)". Exactly how is sensitivity derived from slope? Same problem for c. I am sure this is totally reasonable and straightforward, but it requires an explicit explanation of the derivation.

e) In 'd', "some SEMS were smaller than the marker size" -- In fact, most are smaller than the marker size, leading readers to think they have been omitted. I recommend plotting SEMs in black, ON TOP of the symbols rather than underneath them, so that they all show up.

3. Time course:

The current analysis of onset time may be over-simplistic.

The authors state, in connection with Fig. 4, that the visual response is delayed by 50 ms after ketamine. Actually, the literal text is ambiguous (a dangling phrase) -- it is unclear what exactly is delayed by 50 ms (the visual stimulus onset? The part of the response that occurs 200 to 400 ms after the visual stimulus onset?). More generally, you do not demonstrate that there is a 50 ms lag in the response until figure 5, so this phrase is confusing to the reader. Even there, it is not exactly clear where the 50 ms value comes from.

Neuronal latency measurement (521-523) is hard to assess. Minimally, a line should be added to figs. 5a, 5b, and supplemental 5 a & b to show where the 3 SD criterion falls. However, session-averaged result will be quite different from the mean of the single session results. Would a supplemental figure be helpful?

The stimulus-related response has two separate phases (Figs 4a,b and 5a). Colored traces (left, Fig. 4A) begin to fan out at ~160 ms after stimulus onset, and a 2nd phase of fan-out begins at ~225 ms; there are 2 corresponding inflection points in the blue trace in Fig. 5a (left) -- small rise starting at 160, steep rise starting at 225 ms. While the authors describe ketamine as causing a 50 ms delay, one could also argue for a complete loss of the 1st phase, and a decreased slope but unchanged latency for the 2nd phase.

At risk of being too prescriptive, I suggest that, to test this idea, one might try fitting a series of piecewise linear models to the variance data, with 0, 1, 2 or 3 inflection points, and use AIC to select the most appropriate model. The depth discrimination data look somewhat different, with just a single phase. The 'pre' data of the saline controls show just the reverse pattern -- one phase in the direction-discrimination task and two phases in the depth-discrimination task. This might mean that

there are two phases in both tasks (with earlier latencies for 'depth'), but the first phase is relatively weak and so does not always appear clearly. (Note that shorter latency for the depth task would be consistent with it being the dominant task, less susceptible to interference.)

"Neuronal data analysis" (line 504 and on) states that 2 different time windows are used to measure slope for pre- versus post-ketamine. This seems arbitrary. Did the pattern of results depend on this choice?

Fig. 5C-e: It seems that the motion and depth tasks are combined here. Coding them separately (e.g., with two different colors, other than blue/gold) would be helpful.

Line 265: "We found ... delayed onset of evidence accumulation" -- but not a delay to the peak firing rate or time of evidence accumulation (Fig. 5d,e). This seems surprising, but does not seem to be touched on in the text. Is the build-up faster after ketamine? If not, how else can this be explained?

4. Task switching. The authors state that "asymmetries occur when switching between tasks" but that "We could not find any behavioral evidence for task switching asymmetries in the current task." (No supporting data are shown, however.) But the main point of the task-switching literature is not that there are asymmetries in switch trials (which may or may not occur, depending on the paradigm; I would certainly add the word "may" in 'asymmetries occur', and I would cite Allport, Styles and Hsieh (1994), or Yeung & Monsell 2003 (J. Exp Psych), or an earlier paper -- one of the many that studied this phenomenon in detail). Rather, the main point of the task-switching literature is that there is a difference between switch trials and repeat trials -- this in fact is what is normally meant as a "switch cost". Was this cost examined?

I strongly recommend a figure, perhaps akin to Figure 2, showing switch versus repetition performance (RT and error rate), minimally for the 'pre' data. (More power could be obtained by including both pre-ketamine and pre-saline data sets). A second figure for the 'post' data could be omitted and replaced with a simple statement if there are no pre- or post-ketamine effects. If there *are* switch effects, then of course more analysis would be indicated.

(In a standard switch task, the task cue is removed before the stimulus is presented. In the present paradigm, it remains on the screen through the entire trial. Since the current task is emphasizing task interference and not task costs, this is not a problem. However, if there really are no switch costs, then the timing of the task cue should be noted in the discussion as the most likely reason for why this is.)

Minor details:

line 69: "On the other hand" is confusing. I think the contrast here is with task-specific versus non-task specific effects. This should be made explicit, e.g., "Thus we saw evidence for task-specific effects of ketamine in the direction but not depth task. In addition to these task-specific effects, we also saw non-task specific effects. In particular, <lines 70 and on> ..."

line 84, "task relevant and irrelevant" - not defined. See "definitions" above

line 171, "which was delayed by 50 ms" -- see "definitions" above

Figure 1 legend and Fig. 1 b: "Stimulus strength is described in normalized values"; Figure 3a and d -- I could not find a description of the normalization. (In Fig. 1 it has units of %, but in Fig. 3 it is unitless.)

Figure 2: The medians are missing from the 3 of the top row "pre" data boxes.

line 342: stereotaxic AP values would be helpful, since the position of bony landmarks relative to the brain may depend on the age and size of the animal.

State your box plot conventions just once, in one legend.

Reviewer #3 (Remarks to the Author):

Comments on COMMSBIO-21-0952, "The NMDA receptor antagonist ketamine impairs and delays context-dependent decision making in the parietal cortex".

In the present study, the authors report new observations characterizing neurophysiological signals recorded from the lateral intraparietal area (LIP) of 2 macaque monkeys performing decision-making tasks after the administration of ketamine, an NMDA receptor antagonist. Specifically, the authors recorded LIP single neurons while monkeys switched between two tasks that were randomly intermingled during experimental sessions. In the first task, a direction-discrimination task, monkeys had to make a decision and respond with a saccade based on the motion of moving dots. In the second task, a depth-discrimination task, the response was based on the stereoscopic depth of dots.

Both tasks have been studied in the field, as well as the contribution of LIP neurons in decision making processes. The key novel observation is that ketamine administration affects LIP activity related to decision processes, consistent with an impairment of the behavioral performance. After ketamine administration, the authors described a general enhancement of the build-up in the activity of LIP neurons for the irrelevant stimulus and a delayed build-up onset of the activity during the direction-discrimination task. Behaviorally, the authors showed increased reaction times following ketamine administration.

This manuscript reports findings that will be of interest to a wide range of readers and investigators interested in decision-making. Given the unique nature of the data and its overall consistency with previous reports, I judge that the results are interpretable. However, I have some points that the authors should clarify.

Major comments:

1. The general story of the manuscript is that the administration of ketamine increases neural sensitivity to irrelevant information, and hence, impairs the performance on the task which employs

congruent stimuli. However, this phenomenon only happened in the direction-discrimination task. The authors explained this might be due to the biased difficulty of the tasks or the asymmetries in switching between tasks, which is not convincing in several ways. First, according to their behavior (Figure 2), the monkeys didn't show a difference between the two tasks. Second, the behavior also varies among conditions with different stimulus strengths for the same task (figure 3). The average performance may cover up the difference in some conditions. Plotting the performance on depth-discrimination as a function of normalized stimulus strength may help to reveal it. Third, they didn't show any results on monkeys' performance on switching between tasks. Analyzing the effect of the previous trial's type on the current trial's performance may help answer the question.

2. Disappointingly, the sample sizes confused me a lot. The authors didn't report the number of samples (neurons/sessions) in any figure caption. Although we can count in some figures, it is impossible for the others. Besides, the authors used a different number of neurons without specifying the criterium. 1) They recorded 44 LIP neurons but only plotted 22 neurons in Figure 4 (lines 154-156). Did the authors perform a selection of neurons? If so, please specify the criterium. Also, did the authors use those 22 neurons for all subsequent analyses or all 44 neurons? 2) They varied the motion coherence and binocular correlation among different conditions for different neurons (lines 424 -427). But we can't tell it from the figures. 3) In Figures 5c-b, why the number of samples does not match? Are there a different number of neurons used in these analyses? Please report it.

Minor comments:

1. I found it was not easy to understand the task design in the Results section since the authors described Figure 2 first. Some important terms, including relevant vs. irrelevant information, congruent vs. incongruent stimuli, null stimulus, are only defined in the Methods section or were not defined. Adding an additional paragraph describing the task design at the beginning of the Results section may help the readers understand the entire study more easily.

2. More details about the behavior should be reported, including how many sessions were used? How many trials were included in each session/block?

3. In the behavioral results (lines 69-76), the authors should write clearly that ketamine does not affect RT in Monkey M for the incongruent stimulus during the Depth-discrimination task. An inconsistency between monkeys should be made clear. In the same paragraph, could the authors also add the mean RTs for all conditions along with p-values while describing the results?

4. Figure 1. In the figure legend, please define the abbreviations RF and RT.

5. Figure 4. lines 159-164. "After ketamine administration, the baseline firing rate modestly increased (Pre: 19.7 ± 16.7 spikes/s, Post: 22.2 ± 17.6 spikes/s, Wilcoxon signed-rank test $p = 0.19$), but the stimulus strength-dependent change in firing rate remained unchanged (Figure 4a right).

However, for the irrelevant stimulus, the stimulus strength-dependent change in firing rate increased after ketamine administration, although the firing rate was only minimally affected by stimulus strength before ketamine administration (Figure 4b)." The statistic is provided only for the baseline period. Is a statistic supporting the modulation of the stimulus strength available?

6. Figure 6. I cannot get a good sense of the neuronal response in MT from the analyses of neuronal sensitivity or variance. I would recommend adding the mean response of the MT neurons, similar to what they did in Figures 4a-b.

7. Why was the time window for computing the build-up slope delayed by 50 ms after ketamine administration? How did the author select 50 ms? Is 50 ms the average difference in the RT before and after ketamine administration for both tasks? If that value is not due to a different RT it may be due to a different build-up onset, but from table 2, the build-up onset is delayed by about 65ms.

8. It would be interesting to know whether the authors found evidence of a dose-related effect of ketamine administration, as they used different doses of ketamine.

Reviewer 1

We thank the reviewer for his/her thoughtful comments. In accordance with the review, we added description in the Abstract, performed reanalysis using data from which ketamine presumably got into the brain, and added detailed descriptions concerning the task switching paradigm. Below is our point-to-point reply to the reviewer's comments.

1. The abstract part looks too short, and didn't include their main findings point by point.

We agree with the reviewer that the abstract is too short. We added detailed explanation concerning our main findings point by point.

2. The authors didn't write how long the primate waited between ketamine injection and starting recording. Ketamine need some minutes to get into brain.

We thank the reviewer for this comment. It is known that ketamine can affect neural processing within 5 minutes after administration. Therefore, in this study, we started recording as soon as possible (approximately 10-20 seconds) after administration. As the reviewer pointed out, however, it takes some time for ketamine to diffuse into the brain. To address this concern, we counted the number of fixation errors per minute after ketamine administration, and determined the time when the number of fixation errors statistically increased. Fixation errors significantly increased 3 minutes after ketamine administration (chi-square test, $p < 0.00001$, Figure 7a), and we therefore excluded trials performed in the first 2 minutes after administration. This procedure excluded an average of 21 trials (21 ± 2.7 (mean \pm sd), 5% of the 4 blocks of trials post administration). We reanalyzed all data using the new data set, and confirmed that all the results were reproduced. We revised all figures and added the results from fixation errors in figure 7a. We also added descriptions concerning this point in the methods section as follows.

(lines 532-536) *Fixation error significantly increased after 3 min of ketamine administration (chi-square test, $p < 0.00001$). Therefore, we excluded trials performed in the first 2 min after administration. This procedure excluded an average of 21 trials (21 ± 2.7 [mean \pm SD]), that is 5% of the four blocks of trials post-administration.*

3. The task description is not clear enough to me. Can the authors provide detailed stimuli pattern respective to every black and gray squares in Fig. 1b?

We thank the reviewer for this comment. We revised the explanation concerning the task switching paradigm and the black and gray squares in Fig. 1b in the methods section “Behavioral tasks and training” as follows.

(lines 429-451) *Figure 1a shows an example. The Tin target was positioned in the response field (RF) of the LIP neuron or towards the preferred direction of the MT neuron at 10 degrees eccentricity, whereas the Tout target was positioned diametrically opposite to the Tin target. In this example, the dot stimulus moved upward, the stereoscopic depth was nearer than the fixation point, and Tin was located in the RF of the LIP neuron which was above the fixation point. When the color of the fixation point was magenta, i.e., direction-discrimination task the monkey was required to report the motion direction; thus, the correct choice was the upward target (Tin). Conversely, when the color of the fixation point was cyan, i.e., depth-discrimination task, the monkey was required to report the stereoscopic depth; thus, the correct choice was the downward target (Tout). The monkey had to make a different choice according to the color of the fixation point, even though the presented stimulus was the same. We varied the task difficulty by manipulating the coherence of moving dots (motion coherence) and the binocularly correlated dots (binocular correlation) in the stimulus to examine the parametric effects of the stimulus (Fig. 1b). Motion coherence and binocular correlation were selected according to the table in Figure 1b where positive values indicated coherence/correlation with motion or depth corresponding to the Tin choice, whereas negative values indicated coherence/correlation with motion or depth corresponding to the Tout choice. For example, the top right black square in Figure 1b denotes that the stimulus had a motion coherence of 48% or 96% ($8 \times 6\%$ or $8 \times 12\%$) and a binocular correlation of 48% or 96% ($8 \times 6\%$ or $8 \times 12\%$), and the bottom right gray square denotes that the stimulus had a motion coherence of 48% or 96% ($8 \times 6\%$ or $8 \times 12\%$) and a binocular correlation of -48% or -96% ($-8 \times 6\%$ or $-8 \times 12\%$).*

We added explanation for the congruent and incongruent stimuli in the methods section as follows.

(lines 451-460) *In the example shown in Figure 1a, positive motion coherences correspond to upward motion and negative motion coherences correspond to downward motion. Likewise, positive binocular correlations correspond to the far depth and negative binocular correlations correspond to the near depth. Therefore, the squares in the upper right and lower left (black-colored stimuli) correspond to stimuli where the correct choice was the same for both tasks. Thus, they are referred to as “congruent” stimuli. Conversely, the squares in the upper left and lower right (gray-colored stimuli) correspond to stimuli where monkeys had to make different choices depending on task. Thus, they are referred to as “incongruent” stimuli.*

Reviewer 2

We thank the reviewer for his/her thoughtful comments. In accordance with the review, we added detailed description concerning the stimulus conditions used in this experiments, and revised description and figure concerning figure 3. We also performed further analysis concerning the build-up onset of LIP neurons and behavioral switch cost. Below is our point-to-point reply to the reviewer's comments.

1. Definitions:

Congruent and incongruent stimuli are not defined. Similarly, relevant and irrelevant stimuli are not defined. While the general terms are clear, readers who have not previously encountered dual tasks will be confused. It would help to explicitly state that in both tasks, stimuli have both a depth and a motion, but that depth is irrelevant in the motion task and motion is irrelevant in the depth task.

We agree with the reviewer that the explanation of stimuli was insufficient. We added explanation for the congruent and incongruent stimuli in the methods section as follows. We decided to omit all descriptions of relevant and irrelevant stimuli because these terms are very confusing.

(lines 451-462) *In the example shown in Figure 1a, positive motion coherences correspond to upward motion and negative motion coherences correspond to downward motion. Likewise, positive binocular correlations correspond to the far depth and negative binocular correlations correspond to the near depth. Therefore, the squares in the upper right and lower left (black-colored stimuli) correspond to stimuli where the correct choice was the same for both tasks. Thus, they are referred to as "congruent" stimuli. Conversely, the squares in the upper left and lower right (gray-colored stimuli) correspond to stimuli where monkeys had to make different choices depending on task. Thus, they are referred to as "incongruent" stimuli. For the red-colored stimulus conditions, stimuli were neutral in that they contained either motion or depth information (i.e., the stimulus strength for motion direction and/or depth was zero).*

2. Figure 3: This critical figure is very unclear.

a) In panel a, the blue dashed lines are not visible.

We agree with the reviewer. The blue dashed lines are now visible.

b) Why is choice accuracy for the relevant stimulus identical pre- and post-ketamine? I thought

the message is that, post-ketamine, there is interference from the irrelevant stimulus that degrades performance? Why don't we see this in either animal?

We thank the reviewer for this comment. Because the previous figure was confusing, we replotted them in a different manner. We made it clear that when the monkeys were discriminating motion direction, behavioral sensitivity for motion information was identical pre and post ketamine, but that for binocular correlation was enhanced post ketamine.

c) I was very confused by how "choice bias directed by the irrelevant stimulus feature" could be plotted in terms of choice accuracy. Now I think I see that perhaps there are two different Y axes, one for the dashed lines (accuracy) and one for the solid lines (bias). However, this still does not explain what is actually being plotted. Explicit definitions for "accuracy" and "bias" would be very helpful.

We thank the reviewer for this comment. As described above, we replotted the data so that the y axis is the same for all figures. We revised figure 3a legend as follows.

(legend of figure 3a) *Psychometric function before (blue) and after (yellow) ketamine administration for each monkey. Proportion T_{in} choice is plotted as a function of motion coherence. Data were combined across 13 experiments for monkey K and 18 experiments for monkey M. Lines denote logistic regression fits, and error bars indicate 95% confidence interval.*

d) Panel b plots sensitivity, which we are told is "derived from the slope ... in (a)". Exactly how is sensitivity derived from slope? Same problem for c. I am sure this is totally reasonable and straightforward, but it requires an explicit explanation of the derivation.

We thank the reviewer for this comment. To calculate behavioral sensitivity, we fitted logistic regression as described in the methods section "Behavioral data analysis". Sensitivity to motion coherence and binocular correlation indicates the value of β_1 or β_2 , respectively. We added further description about this point described as follows.

(lines 554-555) *We used β_1 and β_2 as indices for behavioral sensitivity to each stimulus feature.*

e) In 'd', "some SEMS were smaller than the marker size" -- In fact, most are smaller than the marker size, leading readers to think they have been omitted. I recommend plotting SEMs in black, ON TOP of the symbols rather than underneath them, so that they all show up.

We thank the reviewer for helpful comment. We revised the color and format of SEMs in figure 3d, as was pointed out.

3. Time course:

The current analysis of onset time may be over-simplistic.

We thank the reviewer for this comment. We added further analysis and revised the figures as well as the explanations as follows.

3-a) The authors state, in connection with Fig. 4, that the visual response is delayed by 50 ms after ketamine. Actually, the literal text is ambiguous (a dangling phrase) -- it is unclear what exactly is delayed by 50 ms (the visual stimulus onset? The part of the response that occurs 200 to 400 ms after the visual stimulus onset?). More generally, you do not demonstrate that there is a 50 ms lag in the response until figure 5, so this phrase is confusing to the reader. Even there, it is not exactly clear where the 50 ms value comes from.

We thank the reviewer for this comment. We agree with the reviewer that the description for the 50ms lag in the neural analysis was insufficient. We delayed the time window 50ms after ketamine administration, because reaction time was delayed by 50ms with ketamine administration (described in lines 142-145). The drift diffusion model parameter dtR confirmed these results. The 50ms delay is not a delay in visual response (as is observed in area MT), but rather a delay in build-up onset, which reflects the onset of evidence accumulation. We added explanation for this point in the results and methods sections as follows.

(lines 174-180) *Using this mean-subtracted firing rate, we computed the build-up slopes by fitting lines to the responses from 200 to 400 ms after visual stimulus onset. This time window was delayed by 50 ms after ketamine administration because reaction time was delayed by 50 ms and the drift diffusion model parameter dtR was approximately 50 ms, presumably reflecting a 50 ms delay in build-up onset, which was confirmed in the analysis described in the following section.*

(lines 593-598) *We estimated the build-up slope using linear regression using the response from 200 ms to 400 ms after visual stimulus onset to estimate the build-up slope. Because the RT was delayed by approximately 50 ms and the drift diffusion model parameter dtR was approximately 50 ms with ketamine administration, this time window was delayed by 50 ms after ketamine administration for both tasks assuming that build-up onset was delayed by 50 ms.*

We also confirmed the results using identical time windows (no delay) as shown in supplementary figure 7. We added explanation for this point in the results section as follows.

(lines 184-185) *These results were confirmed using identical time windows (without the 50 ms delay) as shown in Supplementary Figure 7.*

3-b) Neuronal latency measurement (521-523) is hard to assess. Minimally, a line should be added to figs. 5a, 5b, and supplemental 5 a & b to show where the 3 SD criterion falls. However, session-averaged result will be quite different from the mean of the single session results. Would a supplemental figure be helpful?

We thank the reviewer for this thoughtful comment. In the previous figure 5a, we demonstrated the time-course of variance calculated from the population waveform shown in figure 4a. Thus, as the reviewer acknowledged, describing the threshold in this figure would be misleading. Therefore, we changed figure 5a to show the average variance across experiments along with their SEMs (shaded areas) in addition to the average threshold across experiments (horizontal dotted lines). We added explanation for this point in the legend of figure 5a as follows.

(legend of figure 5a) *Build-up onset of LIP neurons before and after ketamine administration for the direction-discrimination task (left) and the depth-discrimination task (right). Solid lines denote firing rate variance among different motion coherences at zero binocular correlation for the direction-discrimination task, and among different binocular correlations at zero motion coherence for the depth-discrimination task aligned to visual stimulus onset. Dashed-horizontal lines denote average threshold corresponding to three standard deviations above the baseline. Blue denotes before administration, and yellow denotes after administration. Shaded area denotes SEM.*

Although, in figure 5b, the thresholds were different between the pre and post conditions, we confirmed the results using a single threshold calculated from the baselines of both the pre and the post conditions (Supplementary figure 9). We added explanation for this point in the results section as follows.

(lines 214-216) *Although we used separate thresholds before and after ketamine administration, we confirmed these results when a single threshold was calculated from the baselines of both the pre- and post-conditions (Supplementary Fig. 9).*

3-c) The stimulus-related response has two separate phases (Figs 4a,b and 5a). Colored traces (left, Fig. 4A) begin to fan out at ~160 ms after stimulus onset, and a 2nd phase of fan-out begins at ~225 ms; there are 2 corresponding inflection points in the blue trace in Fig. 5a (left) -- small rise starting at 160, steep rise starting at 225 ms. While the authors describe ketamine as causing a 50 ms delay, one could also argue the for a complete loss of the 1st phase, and a decreased slope but unchanged latency for the 2nd phase.

At risk of being too prescriptive, I suggest that, to test this idea, one might trying fitting a series of piecewise linear models to the variance data, with 0, 1, 2 or 3 inflection points, and use AIC to select the most appropriate model. The depth discrimination data look somewhat different, with just a single phase. The 'pre' data of the saline controls show just the reverse pattern -- one phase in the direction-discrimination task and two phases in the depth-discrimination task. This might mean that there are two phases in both tasks (with earlier latencies for 'depth'), but the first phase is relatively weak and so does not always appear clearly. (Note that shorter latency for the depth task would be consistent with it being the dominant task, less susceptible to interference.)

We thank the reviewer for this thoughtful comment. We attempted to fit piecewise linear models with 1, 2 or 3 inflection points to the variance traces, and calculated AIC to select the most appropriate model. As the reviewer pointed out, models with more than 2 inflection points had lower AICs than a model with 1 inflection point, suggesting that the variance traces had more than 2 phases. However, AICs became lower the more inflection points were added, presumably because the data were noisy. Thus, we could not determine the best number of inflection points, and therefore abandoned this analysis.

The reviewer suggested that the delayed onset could be explained by a loss of the 1st phase and a decrement of the 2nd phase slope. This idea was not supported by the data because, first, the median onset time pre-administration was around 250ms after visual stimulus onset (Figure 5b). Thus, onset time was later than the 1st phase even before ketamine administration. Second, we confirmed that the slope of the variance traces around onset time was not modulated by ketamine. We calculated the slope from 30ms before to 30ms after onset time, and found that they did not differ before and after ketamine administration (Figure 5c). Furthermore, modulation of the slope by ketamine did not correlate with the reaction time delay (Figure 5e). We added these descriptions in the results section "Effects of ketamine on the onset of build-up activity of LIP neurons" as follows.

(lines 211-214) There were no differences in the slope of the variance traces around onset time (calculated from 30 ms before to 30 ms after onset time), indicating that the delayed onset was not due to changes in the slope of the variance traces with ketamine.

(lines 223-228) *A significant correlation was observed between the delay in RT and the delay in build-up onset (Spearman's $r = 0.39$, $p = 0.032$), whereas no correlation was detected between the delay in RT and the difference in onset slope (Spearman's $r = -0.18$, $p = 0.33$), the difference in peak firing rate (Spearman's $r = -0.05$, $p = 0.78$), and the difference in the time from build-up peak to saccade onset (Spearman's, $r = -0.18$, $p = 0.34$).*

3-d) "Neuronal data analysis" (line 504 and on) states that 2 different time windows are used to measure slope for pre- versus post-ketamine. This seems arbitrary. Did the pattern of results depend on this choice?

We thank the reviewer for this comment. We addressed this in comment 3-a).

3-e) Fig. 5C-e: It seems that the motion and depth tasks are combined here. Coding them separately (e.g., with two different colors, other than blue/gold) would be helpful.

We thank the reviewer for this helpful comment. The marker color is now magenta for the direction-discrimination task and cyan for the depth-discrimination task in figure 5.

3-f) Line 265: "We found ... delayed onset of evidence accumulation" -- but not a delay to the peak firing rate or time of evidence accumulation (Fig. 5d,e). This seems surprising, but does not seem to be touched on in the text. Is the build-up faster after ketamine? If not, how else can this be explained?

We thank the reviewer for this comment. We think this is a misunderstanding of the results. We found no change in the peak firing rate and the time from the peak to saccade onset. We did not directly measure the time from stimulus onset to peak, but this was presumably delayed because reaction times were delayed by approximately 50ms and the time from the peak to saccade onset did not change. Because both accumulation onset and reaction times were delayed by approximately 50ms, the time from accumulation onset to peak time (which corresponds to the time of evidence accumulation) was presumably preserved, consistent with finding that build up *per se* did not change. We carefully revised the text so as not to give the impression that there was no delay from stimulus onset to peak time.

4-a) Task switching. The authors state that "asymmetries occur when switching between tasks" but that "We could not find any behavioral evidence for task switching asymmetries in the current task." (No supporting data are shown, however.) But the main point of the task-switching

literature is not that there are asymmetries in switch trials (which may or may not occur, depending on the paradigm; I would certainly add the word "may" in 'asymmetries occur', and I would cite Allport, Styles and Hsieh (1994), or Yeung & Monsell 2003 (J. Exp Psych), or an earlier paper -- one of the many that studied this phenomenon in detail). Rather, the main point of the task-switching literature is that there is a difference between switch trials and repeat trials -- this in fact is what is normally meant as a "switch cost". Was this cost examined?

We thank the reviewer for this comment. As the reviewer pointed out, switch cost is of great importance for evaluating task-switching performance, but it is somewhat difficult to observe switch cost in monkey compared with human. Stoet and Snyder (2003) reported that monkeys did not show switch cost in a paradigm where humans demonstrated switch cost. The switch cost for monkeys was only observed under conditions where inter-trial intervals were very short, suggesting that the switch cost was relatively small compared with humans. As is the case with this report, we could not observe any difference between switch and repetition trials (Supplementary figure 6). We believe that this was because the time from saccade to visual stimulus onset of the next trial was considerably long (more than 2.5sec, median: 3.80sec).

In addition to inserting "may" in 'asymmetries occur', and citing Allport, Styles and Hsieh (1994) and Yeung & Monsell 2003 (J. Exp Psych), we added description of switch cost in supplementary figure 6, along with a detailed description for this point in the discussion section as follows.

(lines 322-330) *Our results were asymmetric in that increased sensitivity to irrelevant information was only observed in the direction-discrimination task. This may be due to asymmetries in switching between tasks. Although we attempted to equate the difficulty of the two tasks, asymmetries may occur when switching between tasks; that is, it is easier to switch from one task to the other than vice versa (19– 21). Asymmetries can be found in switch costs (19, 20); however, we did not observe them in our study (Supplementary Fig. 6). This is presumably because the intertrial interval was too long to observe the switch cost. If the intertrial interval was very short, we may observe asymmetric switch costs as previously reported (3, 21).*

4-b) I strongly recommend a figure, perhaps akin to Figure 2, showing switch versus repetition performance (RT and error rate), minimally for the 'pre' data. (More power could be obtained by including both pre-ketamine and pre-saline data sets). A second figure for the 'post' data could be omitted and replaced with a simple statement if there are no pre- or post-ketamine effects. If there *are* switch effects, then of course more analysis would be indicated.

(In a standard switch task, the task cue is removed before the stimulus is presented. In the present paradigm, it remains on the screen through the entire trial. Since the current task is emphasizing

task interference and not task costs, this is not a problem. However, if there really are no switch costs, then the timing of the task cue should be noted in the discussion as the most likely reason for why this is.)

As described above in the response to the **4-a)** comment, we compared behavioral performance (Error rate and RT) between switch versus repetition trials using both pre-ketamine and pre-saline data sets (supplementary figure 6). We did not find any difference in switch cost for both pre- and post- ketamine conditions.

As the reviewer pointed out, we presented the task cue on the screen throughout the entire trial, and this may be part of the reason why we did not observe switch costs. However, we believe that the long inter-trial interval had a bigger effect on the lack of switch costs.

Minor details:

1) line 69: "On the other hand" is confusing. I think the contrast here is with task-specific versus non-task specific effects. This should be made explicit, e.g., "Thus we saw evidence for task-specific effects of ketamine in the direction but not depth task. In addition to these task-specific effects, we also saw non-task specific effects. In particular, ..."

We thank the reviewer for this comment. We revised text described as bellow.

(lines 84-87) *These results indicate that ketamine impaired decision making for the incongruent stimulus only during the direction-discrimination task in a task-specific manner. In addition to these task-specific effects, we also observed a task-independent effect on the RT.*

2) line 84, "task relevant and irrelevant" - not defined. See "definitions" above

As we explained in major comment **1)**, we decided to omit all descriptions of relevant and irrelevant stimuli because these terms are very confusing.

3) line 171, "which was delayed by 50 ms" -- see "definitions" above

This is addressed in major comment **3-a)**.

4) Figure 1 legend and Fig. 1 b: "Stimulus strength is described in normalized values"; Figure 3a and d -- I could not find a description of the normalization. (In Fig. 1 it has units of %, but in Fig. 3 it is unitless.)

We thank the reviewer for this comment. We revised all figures so that stimulus strength is described in either motion coherence (%) or binocular correlation (%).

5) *Figure 2: The medians are missing from the 3 of the top row "pre" data boxes.*

We thank the reviewer for this comment. We changed the color of the median to magenta for visibility.

6) *line 342: stereotaxic AP values would be helpful, since the position of bony landmarks relative to the brain may depend on the age and size of the animal.*

We agree with the reviewer on this point, but we did not set the recording chambers based on precise stereotaxic AP values. As described in the methods section “Subjects and surgery” (lines 390-393), we mounted the recording chamber according to a previous study (Rao et al., 2012). Furthermore, we identified the MT and LIP areas from their physiological response properties on the day of the experiment. These are typical methods for identifying the two areas.

7) *State your box plot conventions just once, in one legend.*

We agree with the reviewer on this point. We revised the legend accordingly.

Reviewer 3

We thank the reviewer for his/her thoughtful comments. In accordance with the review, we added comments and further analysis concerning the asymmetries of the ketamine effect, and detailed description concerning sample size of experiments, stimulus conditions, and procedures of each analysis. Below is our point-to-point reply to the reviewer's comments.

Major comments:

1. The general story of the manuscript is that the administration of ketamine increases neural sensitivity to irrelevant information, and hence, impairs the performance on the task which employs congruent stimuli. However, this phenomenon only happened in the direction-discrimination task. The authors explained this might be due to the biased difficulty of the tasks or the asymmetries in switching between tasks, which is not convincing in several ways. First, according to their behavior (Figure 2), the monkeys didn't show a difference between the two tasks. Second, the behavior also varies among conditions with different stimulus strengths for the same task (figure 3). The average performance may cover up the difference in some conditions. Plotting the performance on depth-discrimination as a function of normalized stimulus strength may help to reveal it. Third, they didn't show any results on monkeys' performance on switching between tasks. Analyzing the effect of the previous trial's type on the current trial's performance may help answer the question.

We thank the reviewer for this comment. First, we agree that the monkeys did not show a difference between the two tasks and does not explain the asymmetry. Second, the performance on depth discrimination as a function of normalized stimulus strength is shown in supplementary figure 2, and this also does not explain the asymmetry. Third, we newly analyzed the effect of the previous trial. As shown in supplementary figure 6, we could not find any behavioral evidence for task switching asymmetries. This can be attributed to the task setting because it is known that switch costs are only observed when inter-trial intervals are very short. In our experiments, the time from saccade to visual stimulus onset of the next trial was considerably long (more than 2.5sec, median: 3.80sec).

Since we could not find any evidence that explains the asymmetry between the two tasks from behavioral analysis, we further analyzed the LIP responses and found that the timing of accumulation of irrelevant information was asymmetric. In supplementary figure 8, we plotted firing rate variance among stimulus conditions when irrelevant information was accumulated using all data from pre-administration trials (n=44; saline: 22, ketamine: 22). Accumulation of motion information during depth discrimination was apparent around 200ms, whereas accumulation of depth information during direction discrimination was only apparent after 500ms. The accumulation of depth

information during direction discrimination became apparent after 200ms after ketamine administration. These results suggest that the monkeys were successful in suppressing irrelevant depth information for a few hundreds of milliseconds, but that was impaired with ketamine administration.

We added a new section “Asymmetry of ketamine effects” concerning this point in the results, and added explanations in discussion section as follows.

(lines 321322-329330) *Our results were asymmetric in that increased sensitivity to irrelevant information was only observed in the direction-discrimination task. This may be due to asymmetries in switching between tasks. Although we attempted to equate the difficulty of the two tasks, asymmetries may occur when switching between tasks; that is, it is easier to switch from one task to the other than vice versa (19- 21). Asymmetries can be found in switch costs (19, 20); however, we did not observe them in our study (Supplementary Fig. 6). This is presumably because the intertrial interval was too long to observe the switch cost. If the intertrial interval was very short, we may observe asymmetric switch costs as previously reported (3, 21).*

(lines 3301-335336) *We further investigated asymmetries in LIP responses and found a task-dependent difference in the way irrelevant information was accumulated. This asymmetry might be attributed to the order of task training. We trained the monkeys to discriminate motion direction first, and subsequently trained them to discriminate stereoscopic depth. Whether the order of training actually contributes to the way irrelevant information is accumulated should be investigated in future research.*

2. Disappointingly, the sample sizes confused me a lot. The authors didn't report the number of samples (neurons/sessions) in any figure caption. Although we can count in some figures, it is impossible for the others. Besides, the authors used a different number of neurons without specifying the criterium.

We thank the reviewer for this comment. We absolutely agree with the reviewer that the description about the number of the data is difficult to understand. We added detailed explanation in the text and figures described as follows point by point, and also revised the figures so that the data distributions are more visible.

2-1) They recorded 44 LIP neurons but only plotted 22 neurons in Figure 4 (lines 154-156). Did the authors perform a selection of neurons? If so, please specify the criterium. Also, did the

authors use those 22 neurons for all subsequent analyses or all 44 neurons?

As described in methods section “Experimental protocols” (lines 511-516), we recorded a total of 44 LIP neurons. Of these, 22 were administered with saline and the other 22 were administered with ketamine. Therefore, in the figure 4 which describes ketamine administration, we did not exclude any data samples. We added further explanation as follows.

(lines 511-516) *We recorded extracellular activity from 44 isolated neurons in the LIP area with 22 saline administration (Monkey K: 10 neurons, Monkey M: 12 neurons) and 22 ketamine administration (Monkey K: 10 neurons, Monkey M: 12 neurons) experiments. We also recorded from 23 isolated neurons in the MT area with 14 saline administration (Monkey K: 5 neurons, Monkey M: 9 neurons) and 9 ketamine administration (Monkey K: 3 neurons, Monkey M: 6 neurons) experiments.*

2-2) They varied the motion coherence and binocular correlation among different conditions for different neurons (lines 424 -427). But we can't tell it from the figures.

We thank the reviewer for this comment. We revised all figures so that stimulus strength is described in either motion coherence (%) or binocular correlation (%). However, because all figures describe population averages, motion coherence and binocular correlation are described as (x 6% or 12%).

2-3) In Figures 5c-b, why the number of samples does not match? Are there a different number of neurons used in these analyses? Please report it.

We are sorry for the lack of explanation on this point. In the onset analysis, we determined onset by setting a threshold corresponding to three SDs above baseline. However, there were some data samples where we were unable to detect the onset time because they did not demonstrate typical buildup (This is not strange as reported in *Meister et al., 2013*. They reported that the degree of buildup is heterogeneous). Thus, onset time is described using detectable data sets.

In our new analysis, we focused on data sets with definite onset time (16 out of 22 for the direction discrimination task and 15 out of 22 for the depth discrimination task). Furthermore, we changed the beginning of the baseline time window from 300ms to 200ms before stimulus onset. This is because the saccade targets were presented 300ms before stimulus onset, and LIP neurons responded to them. Thus, we prevented the involvement of LIP response to visual target presentation.

We added detailed description as follows in the methods section “Neuronal data analysis”.

(lines 607-620) *We quantified the onset of build-up activity by computing the variance of the firing rate among responses to different motion coherences for the direction- discrimination task and different binocular correlations for the depth-discrimination task as previously reported (15). To determine when the variance deviated from baseline, we set a threshold corresponding to three SDs above baseline, defined as the average variance from 200 ms before to visual stimulus onset. There were some data samples in which we could not determine the onset time because the variance did not deviate from baseline. Thus, for onset analysis with saline administration, we excluded 7 and 8 data samples for the direction-discrimination task and depth-discrimination task, respectively, and with ketamine administration, we excluded 6 and 7 data samples, respectively. With these data sets, we examined the build-up peak firing rate, and the time from build-up peak to saccade onset using the response for preferred choices from 200 ms before to the saccade time. All analyses were performed using two-sided non-parametric statistical tests (Wilcoxon signed-rank test).*

Minor comments:

1. I found it was not easy to understand the task design in the Results section since the authors described Figure 2 first. Some important terms, including relevant vs. irrelevant information, congruent vs. incongruent stimuli, null stimulus, are only defined in the Methods section or were not defined. Adding an additional paragraph describing the task design at the beginning of the Results section may help the readers understand the entire study more easily.

We thank the reviewer for this comment. We added description for the congruent vs. incongruent stimulus in the method section “Behavioral tasks and training”. Furthermore, we added an additional paragraph describing the task design at the beginning of the Results section as follows. We also abandoned the description of relevant vs irrelevant stimuli, as well as null stimulus because these were confusing.

(lines 451-462) *In the example shown in Figure 1a, positive motion coherences correspond to upward motion and negative motion coherences correspond to downward motion. Likewise, positive binocular correlations correspond to the far depth and negative binocular correlations correspond to the near depth. Therefore, the squares in the upper right and lower left (black-colored stimuli) correspond to stimuli where the correct choice was the same for both tasks. Thus, they are referred to as “congruent” stimuli. Conversely, the squares in the upper left and lower right (gray-colored stimuli) correspond to stimuli where monkeys had to make different choices depending on task. Thus, they are referred to as “incongruent” stimuli. For the red-colored stimulus conditions, stimuli were neutral in that they contained either motion or depth information (i.e., the stimulus strength for motion direction and/or depth was zero).*

(lines 62-74) *The monkeys performed a cued task-switching paradigm where they were instructed to discriminate either motion direction (direction-discrimination task) or stereoscopic depth (depth-discrimination task) contained in a moving random-dot stereogram depending on the color of the fixation point (Fig. 1). Task difficulty varied by changing the percentage of dots that moved in a particular direction (motion coherence) or fell in a particular depth plane. Visual stimuli consisted of three conditions: congruent stimuli where the correct choice was the same for both tasks, incongruent stimuli where the correct choice was different depending on the task, and neutral stimuli where at least one of the two stimulus dimensions contained no information (0% motion coherence and/or 0% binocular correlation). Here, changes in motion coherence were relevant for the direction-discrimination task but were irrelevant for the depth-discrimination task. Conversely, changes in binocular correlation were relevant for the depth-discrimination task but were irrelevant for the direction-discrimination task.*

2. More details about the behavior should be reported, including how many sessions were used? How many trials were included in each session/block?

We thank the reviewer for this comment. We already described the number of blocks before and after administration in methods section “Experimental protocol” as follows.

(lines 522-523) *We collected data for four blocks, which lasted about 30 min, before ketamine or saline administration.*

We did not describe the number of trials in each block. We added description about the number of trials in a block in the methods section “Behavioral tasks and training” as follows.

(lines 462-465) *Neutral (red-colored) conditions contained 36 trials, and congruent (black-colored) and incongruent (gray-colored) conditions contained 8 trials each in a block. Therefore, a total of 52 trials were used for each of the two tasks, yielding 104 trials within a block.*

For behavioral analysis, we used all sessions. This includes both LIP and MT recording sessions for both saline and ketamine administration. We added further description about this point in methods section “Behavioral data analysis” as follows.

(lines 539-540) *We used 36 saline and 31 ketamine administration experiments covering both LIP and MT recordings for behavioral analysis.*

3. In the behavioral results (lines 69-76), the authors should write clearly that ketamine does not affect RT in Monkey M for the incongruent stimulus during the Depth-discrimination task. An inconsistency between monkeys should be made clear. In the same paragraph, could the authors also add the mean RTs for all conditions along with p-values while describing the results?

When we reanalyzed our data excluding trials right after ketamine administration (a comment from reviewer 1), we found statistical difference of RT in monkey M for the incongruent stimulus for the depth-discrimination task. We summarized the median RTs and the error proportion along with the p-values in table 1.

4. Figure 1. In the figure legend, please define the abbreviations RF and RT.

We thank the reviewer for this comment. We added description about abbreviations RF and RT in the legend of figure 1 as follows.

(legend of figure 1) RF: response field of LIP neuron, RT: reaction time.

5. Figure 4. lines 159-164. "After ketamine administration, the baseline firing rate modestly increased (Pre: 19.7 ± 16.7 spikes/s, Post: 22.2 ± 17.6 spikes/s, Wilcoxon signed-rank test $p = 0.19$), but the stimulus strength-dependent change in firing rate remained unchanged (Figure 4a right). However, for the irrelevant stimulus, the stimulus strength-dependent change in firing rate increased after ketamine administration, although the firing rate was only minimally affected by stimulus strength before ketamine administration (Figure 4b)." The statistic is provided only for the baseline period. Is a statistic supporting the modulation of the stimulus strength available?

We thank the reviewer for this comment. In order to statistically test the modulation depending on the stimulus strength, we calculated the build-up slope of the population time-course waveform using the response from 200ms to 400ms for pre-administration and 250ms to 450ms for post-administration after visual stimulus onset, and estimated sensitivity to motion coherence or binocular correlation, as calculated in each data set described in the methods section (lines 593-606). We added description about results of the z-test applied to the fitted parameter α_1 and α_3 as follows.

(lines 161-169) Before ketamine administration, the firing rates changed depending on the stimulus strength for relevant information (z-test, $p = 0.0001$). After ketamine administration, the baseline firing rate increased modestly (Pre: 19.7 ± 16.7 spikes/s, Post: 22.2 ± 17.6 spikes/s, Wilcoxon

signed-rank test $p = 0.19$) but the stimulus strength-dependent change in firing rate remained (z -test, $p = 0.0056$). However, the stimulus strength-dependent change in firing rate was only observable after ketamine administration for irrelevant information (z -test, $p = 0.0022$), whereas this was not observed before ketamine administration (z -test, $p = 0.49$).

6. Figure 6. I cannot get a good sense of the neuronal response in MT from the analyses of neuronal sensitivity or variance. I would recommend adding the mean response of the MT neurons, similar to what they did in Figures 4a-b.

We thank the reviewer for this comment. We agree with the reviewer that explanations and figures for the MT analysis were insufficient. We added detailed explanations in methods section “Neuronal data analysis” as follows. We also added a figure on the firing rate time-course so that the MT responses with ketamine administration are easier to understand.

(lines 621-632) *We quantified neural sensitivity to the binocular correlation of MT neurons as previously reported (17). The binocular correlation-dependent increase in firing rate was calculated from 50 ms after visual stimulus onset to saccade onset using the data from stimulus conditions where the motion coherence was fixed at 0 (Fig. 1b). The firing rates for preferred and anti-preferred stimuli were fit using linear regression as follows:*

$$\text{Sensitivity to binocular correlation}_{pref} = \gamma_0 + \gamma_1 \text{Pref stimulus strength}$$

Sensitivity to binocular correlation}_{anti-pref} = \gamma_2 + \gamma_3 \text{Anti - pref stimulus strength},
where γ_0-3 are fitted parameters. Logistic regression functions were fit for the direction-discrimination task before and after ketamine or saline administration. We used γ_1 and γ_3 as indices for sensitivity to binocular correlation for preferred and anti-preferred stimuli, respectively.

7. Why was the time window for computing the build-up slope delayed by 50 ms after ketamine administration? How did the author select 50 ms? Is 50 ms the average difference in the RT before and after ketamine administration for both tasks? If that value is not due to a different RT it may be due to a different build-up onset, but from table 2, the build-up onset is delayed by about 65ms.

We thank the reviewer for this comment. As described in the reply to comment 3-a) of Reviewer 2, we delayed the time window 50ms after ketamine administration, because reaction time was delayed by 50ms with ketamine administration (described in lines 142-145). The drift diffusion model parameter dtR confirmed these results. We added explanation for this point in the results and

methods sections as follows.

(lines 174-180) *Using this mean-subtracted firing rate, we computed the build-up slopes by fitting lines to the responses from 200 to 400 ms after visual stimulus onset. This time window was delayed by 50 ms after ketamine administration because reaction time was delayed by 50 ms and the drift diffusion model parameter dtR was approximately 50 ms, presumably reflecting a 50 ms delay in build-up onset, which was confirmed in the analysis described in the following section.*

(lines 593-598) *We estimated the build-up slope using linear regression using the response from 200 ms to 400 ms after visual stimulus onset to estimate the build-up slope. Because the RT was delayed by approximately 50 ms and the drift diffusion model parameter dtR was approximately 50 ms with ketamine administration, this time window was delayed by 50 ms after ketamine administration for both tasks assuming that build-up onset was delayed by 50 ms.*

We also confirmed the results using identical time windows (no delay) as shown in supplementary figure 7. We added explanation for this point in the results section as follows.

(lines 184-185) *These results were confirmed using identical time windows (without the 50 ms delay) as shown in Supplementary Figure 7.*

8. It would be interesting to know whether the authors found evidence of a dose-related effect of ketamine administration, as they used different doses of ketamine.

We thank the reviewer for this comment. We agree with the reviewer that a dose-related effect of ketamine is an interesting point, but we could not find any evidence in our study. This is presumably because of the small sample sizes for each dosage for each monkey (dosage (number of experiments) Monkey K: 0.25mg/kg (6), 0.35mg/kg (5), and 0.5mg/kg (2), Monkey M: 0.25mg/kg (3), 0.35mg/kg (4), 0.4mg/kg (3), and 0.5mg/kg (8)).

Reviewers' comments:

Reviewer #1 (Remarks to the Author):

Many thanks to the authors for addressing my concerns. All my concerns have been properly solved.

Reviewer #2 (Remarks to the Author):

For the most part, all my concerns have been satisfactorily addressed. I have just a single point of confusion remaining, in connection with original question 3b.

In the text and in Figure 3B, we learn that ketamine increases inappropriate sensitivity to binocular correlation during the direction discrimination task. We would expect that being more sensitive to an inappropriate cue would decrease percentage correct in Fig. 3A -- but this is not the case. Perhaps the increased sensitivity to binocular correlation is just too small a factor to produce a measurable effect on motion discrimination? This seems a reasonable explanation for the data of Monkey M, but the effect of ketamine on binocular sensitivity in Monkey K seems too large for this explanation to be correct. Readers will be confused! Please explain.

Reviewer #3 (Remarks to the Author):

The authors have done an excellent job in the revision and they have addressed all my concerns. I have no further comments.

Reviewer 2

We thank the reviewer for his/her thoughtful comment. In accordance with the review, we have added a detailed explanation regarding Figures 3a and 3b.

In the text and in Figure 3B, we learn that ketamine increases inappropriate sensitivity to binocular correlation during the direction discrimination task. We would expect that being more sensitive to an inappropriate cue would decrease percentage correct in Fig. 3A -- but this is not the case. Perhaps the increased sensitivity to binocular correlation is just too small a factor to produce a measurable effect on motion discrimination? This seems a reasonable explanation for the data of Monkey M, but the effect of ketamine on binocular sensitivity in Monkey K seems too large for this explanation to be correct. Readers will be confused! Please explain.

We thank the reviewer for this comment. This is absolutely one of the most important points of this study, but we admit that we did not explain it well enough. Yes, we would expect that being more sensitive to an inappropriate cue would decrease performance when data are combined across irrelevant stimulus features. However, as we mentioned in “Behavioral data analysis” of the Methods section, we evaluated the behavioral sensitivity to motion coherence in the direction-discrimination task (figure 3a) using stimuli with 0% binocular correlation. Conversely, the sensitivity to binocular correlation in the direction-discrimination task (figure 3b) was evaluated using stimuli with 0% motion coherence. Thus, sensitivity to motion and sensitivity to depth are evaluated independently. We should have clarified that the stimuli used to plot the psychometric functions in figures 3a and 3b were different. Therefore, we added a panel explaining the stimuli used in figures 3a, 3b, and supplementary figures 2a and 2b. We also added the following statement to the Results section and the legends of Figure 3 and Supplementary Figure 2.

(lines 95-100) *To answer this question, we analyzed psychometric functions for neutral stimuli before and after ketamine administration during the direction-discrimination task (Fig. 3). Sensitivity to relevant stimuli was quantified by the slope of the psychometric functions using logistic regression (11, 12) for visual stimuli with 0% binocular correlation (Fig. 3a, right panel). Sensitivity to irrelevant stimuli was quantified with visual stimuli with 0% motion coherence (Fig. 3b, right panel). Prior to ketamine administration, ~*

(Legend of Figure 3) *a) Psychometric function before (blue) and after (yellow) ketamine administration for each monkey. Proportion Tin choice is plotted as a function of motion coherence*

*with 0% binocular correlation. Stimuli used to plot psychometric functions are shown on the right. Data were combined across 13 experiments for monkey K and 18 experiments for monkey M. Lines denote logistic regression fits, and error bars indicate 95% confidence interval. b) Proportion Tin choice is plotted as a function of binocular correlation with 0% motion coherence. Asterisks indicate statistically significant results (Z test; ** $p < 0.01$, * $p < 0.05$). Conventions are the same as in (a).*

(Legend of Supplementary Figure 2) *a) Psychometric function before (blue) and after (yellow) ketamine administration for each monkey. Proportion Tin choice is plotted as a function of binocular correlation with 0% motion coherence. The conventions are the same as in Figure 3.*

REVIEWERS' COMMENTS:

Reviewer #2 (Remarks to the Author):

Thank you for the clarifications.